# **IA2**: ALIGNMENT WITH ICL ACTIVATIONS IMPROVES SUPERVISED FINE-TUNING

**Aayush Mishra, Daniel Khashabi$^\heartsuit$ & Anqi Liu$^\heartsuit$**
Department of Computer Science, Johns Hopkins University

## ABSTRACT

Supervised Fine-Tuning (SFT) is used to specialize model behavior by training weights to produce intended target responses for queries. In contrast, In-Context Learning (ICL) adapts models during inference with instructions or demonstrations in the prompt. ICL can offer better generalizability and more calibrated responses compared to SFT in data scarce settings, at the cost of more inference compute. In this work, we ask the question: *Can ICL's internal computations be used to improve the qualities of SFT?* We first show that ICL and SFT produce distinct activation patterns, indicating that the two methods achieve adaptation through different functional mechanisms. Motivated by this observation and to use ICL's rich functionality, we introduce **I**CL **A**ctivation **A**lignment (**IA2**), a self-distillation technique which aims to replicate ICL's activation patterns in SFT models and incentivizes ICL-like internal reasoning. Performing **IA2** as a priming step before SFT significantly improves the accuracy and calibration of model outputs, as shown by our extensive empirical results on 12 popular benchmarks and two model families. This finding is not only practically useful, but also offers a conceptual window into the inner mechanics of model adaptation.

## 1 INTRODUCTION

LLMs are general purpose models but are often used in specialized applications. For example, a news aggregator app may need to classify articles into predefined categories. A popular approach for adapting LLMs is **Supervised Fine-Tuning (SFT)** which uses a dataset of labeled samples to train and adapt LLMs on narrow downstream tasks. With the power of parameter efficient fine tuning (PEFT) techniques (Hu et al., 2021; Liu et al., 2022), SFT models are often just a small set of parameters which can be efficiently loaded on/off GPU memory making them extremely useful. However, SFT typically requires a large set of labeled samples to generalize well on new tasks (Le Scao & Rush, 2021), which can be expensive to collect.

In contrast, **In-Context Learning (ICL)** (Brown et al., 2020) is used to adapt and steer LLM behavior during inference time. The model is given a query preceded by in-context demonstrations or instructions, which help it "learn" the demonstrated task and answer accordingly. Prior work (Duan et al., 2024) shows (and our experiments concur (§5)) that *ICL generalizes well in a few-shot setting and typically produces well calibrated responses*. However, using ICL comes at a cost. ICL demonstrations/instructions use up valuable context space (increasing the cost of running each query) which could otherwise be used for processing more query and response tokens.

These subtle but important differences between ICL and SFT motivate us to investigate the difference in their functional behavior. We find that while their token space behavior may look similar on the surface, **models produce disparate internal activations under ICL and SFT** (§3). As activations are a footprint of the model's internal processing before it produces the next token, divergent activation patterns highlight a difference in how the two methods achieve adaptation, an idea supported by prior work (Shen et al., 2024). We hypothesize that when the model performs ICL, its activations contain rich information about how to extract generalizable patterns from the context. This information may be absent in SFT models especially in a few-shot setting, where they are prone to shortcut learning/overfitting on target responses. Hence, we ask the research question: *Can the information rich ICL activations be used to improve the quality of SFT?*

---

$^\heartsuit$Equal advising. Correspondence to: {amishr24,danielk,aliu.cs}@jhu.edu

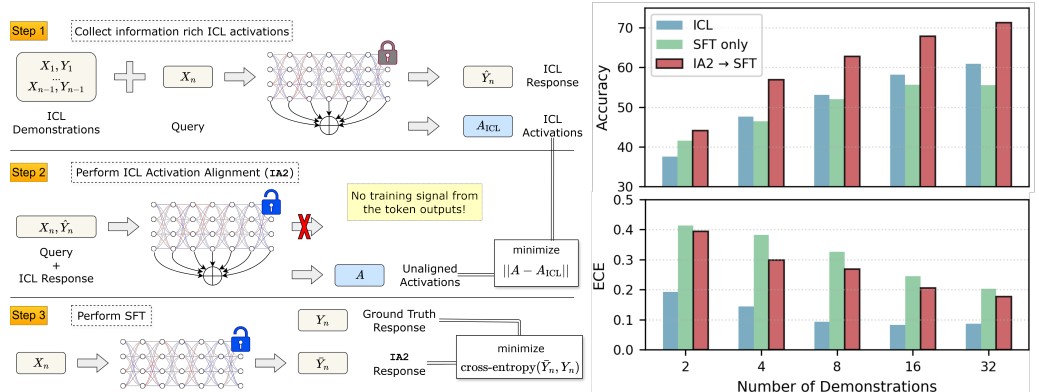

Figure 1: An overview of our improved SFT pipeline. Standard SFT only enforces output space alignment (between the model's response and a target response—only step 3 above), resulting in subpar performance and mis-calibration in low-data settings. In contrast, our method **IA2** enforces functional alignment with ICL by matching the rich activation patterns produced when performing ICL. **IA2** *priming* before SFT boosts the quality of adaptation. We show this improvement through performance comparison charts aggregated across models and datasets. See section 5 for details.

Recent works have tried to distill context in the weights of LLMs (Snell et al., 2022; Chen et al., 2024b), but they use training signals only from response texts, which may suffer from the same SFT issues highlighted above. Simply training a model to reproduce the outputs of an ICL-conditioned model does not ensure that it functions like an ICL-conditioned model.

To address this, we propose **I**CL **A**ctivation **A**lignment (**IA2**), a self-distillation method to enforce alignment with the model's own functional behavior when performing ICL (§4). See Figure 1 for an overview of our proposed method: (1) collect information rich ICL activations and (2) enforce functional alignment with ICL. Then (3) perform SFT on this primed model. We show that **priming models with IA2 before SFT** *using the same data*, **drastically improves the performance** of the adapted model on a variety of text classification and generation tasks (§5). In addition, we show that **IA2 provides an important training signal that is unavailable with SFT only training**. We trained over 13,000 models spanning 12 benchmarks, to validate our findings, which not only signify the practical benefits of **IA2**, but offer a conceptual window into the inner mechanics of model adaptation. In summary:

⋆ We show that ICL and SFT with the same data, do not align in the model's activation space, highlighting a gap in their functional behavior (§3).

⋆ We propose **IA2**—ICL Activation Alignment, to enforce alignment with ICL's functional behavior. This priming step drastically improves the performance of SFT models (see Figure 1, §4).

⋆ We show that the **IA2** training signal is not present in SFT only training, highlighting its importance in improving the quality of adaptation (§5).

## 2 BACKGROUND AND NOTATION

**Transformer Language Models:** A standard decoder-only transformer based LM $M_\Theta$ with parameters $\Theta$ consists of a stack of self-attention (SA) and linear layers. A sequence of token vectors $T = [t_i]_{i=1}^R$ is processed by applying the SA and linear projections on each token at each layer, until the last token projection at the final layer is passed through the LM head to predict the most likely next token in the sequence. The SA operation is special because it is affected by other tokens in the sequence. For a standard SA operation which uses 4 weight matrices, $W_Q, W_K, W_V$ and $W_O \in \Theta$ (we will call it the set $W_{QKVO}$ for brevity), the output at each token position is given by:

$$\text{SA}(T; W_{QKVO}) = Z = \left[ z_i = \sigma\left(\frac{q_i K^T}{\sqrt{d}}\right) V \cdot W_O \right]_{i=1}^R \tag{1}$$

where $\sigma(.)$ denotes Softmax, $q_i = t_i \cdot W_Q$, $K = t_{:i} \cdot W_K$, $V = t_{:i} \cdot W_V$ ($t_{:i}$ means first $i$ tokens). Note that $Z$ has the same shape ($R \times d$) as the input $T$. In this work, we study the interplay of changes in model behavior induced by changing the context $T$ or the weights $W_{QKVO}$.

**Supervised Fine-Tuning:** Given a set $\mathcal{D}_\mathcal{T} = \{(X_i, Y_i)\}_{i=1}^N$ of $N$ input ($X$)–output ($Y$) pairs illustrating a task $\mathcal{T}$, we can fine-tune the model weights $\Theta \to \Theta'$ to produce relevant responses for new inputs (say $X_t$), i.e., $M_{\Theta'}(X_t) = \bar{y}_t$. Here, $\bar{y}_t$ is the first token of the whole response $\bar{Y}_t$ which can be extended by repeated application of $M_{\Theta'}$ on the growing sequence. This process is performed as follows. If the ground truth response $Y_i$ contains $G$ tokens, we generate $G$ new tokens with the model to get $\bar{Y}_i$ and minimize the cross-entropy loss over all generated tokens. The SFT loss is defined as:

$$\mathcal{L}_{\text{SFT}} = \sum_{i=1}^N \sum_{j=1}^G \text{cross-entropy}(\bar{Y}_{ij}, Y_{ij}) \tag{2}$$

**In-Context Learning:** Using $\mathcal{D}_\mathcal{T}$, $M_\Theta$ can also be prompted with the concatenated sequence of demonstration tokens $I = [X_1 \circ Y_1 \circ \ldots \circ X_n \circ Y_n]$ along with a new test sample $X_t$ to get a task-appropriate response without any weight updates. $M_\Theta$ processes the examples in its prompt to understand the task and produces a response $M_\Theta(I \circ X_t) = \hat{y}_t$ accordingly. Remarkably, $\hat{Y}_t$ is often similar to the expected response $Y_t$. Hence, ICL serves as a useful inference-time adaptation method. ICL was first illustrated in GPT-3 (Brown et al., 2020) and is widely used to adapt generic LMs at the the inference-time. Today's frontier models have the ability to follow instructions directly, so a generic instruction could be considered a form of ICL demonstrations. In this work, we will focus on the classic ICL setting with a demonstration set of input-output pairs.

**Activations:** As the SA operation uses every token in the context, it produces different outputs for different sequences of tokens. We denote the hidden SA outputs as activations. Activations are a footprint of the model's internal processing which can be used to study model behavior changes with changing contexts. If $M_\Theta$ consists of $L$ layers, and it processes $R$ tokens, we get an activation tensor $A$ of size $L \times R \times d$. In this work, we study sequences of tokens $T = [I \circ X]$ for ICL; and $T = X$ without ICL under different model weights, where $I$ denotes tokens of ICL demos and $X$ denotes tokens of the query. Here, we also define activation similarity ($\texttt{asim}\,(A_1, A_2)$) as the token-wise cosine similarity between two activation tensors. Note that $\texttt{asim}$ is of size $L \times R$.

## 3 ARE ICL AND SFT PRACTICALLY THE SAME?

Recent studies (Von Oswald et al., 2023; Akyürek et al., 2022) claim that ICL in transformer models works by an internal gradient descent mechanism. They show that the self-attention operation can produce outputs as if the model parameters were updated using ICL demos in context. This implies a functional equivalence between the two. Under this equivalence, we should expect the base model with ICL demos to produce similar activations at the output token positions as the SFT model produces without ICL demos. We investigated whether this phenomenon is actually exhibited by LLMs.

**ICL and SFT produce different activations:** We collected ICL activations $A_{\text{ICL}}$ (using $T = [I \circ X]$) for 100 test samples from multiple datasets (SST2, AGNews, etc.) in a variety of models (Qwen3 (Yang et al., 2025) and Llama-3.2 (Grattafiori et al., 2024) family, 1B$\leftrightarrow$16 layers, 3B$\leftrightarrow$28 layers, and 4B$\leftrightarrow$36 layers). We then trained SFT models using the same ICL demos until convergence, and collected activations $A_{\text{SFT}}$ (using $T = X$). Experiment details can be found in §B. Then, we calculated $\texttt{asim}\,(A_{\text{ICL}}, A_{\text{SFT}})$

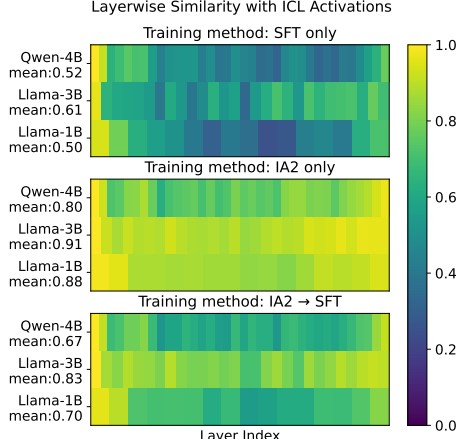

Figure 2: Layerwise Similarity with ICL Activations. SFT models have small $\texttt{asim}$ with ICL. This indicates differing functional behavior for adaptation. Our recommended pipeline (**IA2** $\to$ SFT) aligns much more with ICL than using only SFT, and performs much better as a result (see Figure 1).

at output token positions, and plotted the average across tokens, samples, and datasets (see Figure 2– SFT only in the first row). We see that ICL and SFT activations are not aligned across different models. It is noteworthy that the activations align better near the initial and final layers where we expect the tokens to be processed at an individual level, but are misaligned in the middle where we expect the whole demonstration set to be processed at an abstract level.

**Why should we care about alignment with ICL?** The difference in functional behavior of ICL and SFT is not just hidden in activation patterns. It also surfaces in the form of expected calibration error (Guo et al., 2017) as seen in Figure 1. ICL is much more calibrated in its responses than SFT at similar (slightly better) accuracy levels. We suspect that this is due to the output-oriented nature of the SFT training signal, which can allow it to learn shortcuts that can fail on new data. In contrast, as ICL relies on complex circuits (Elhage et al., 2021) that need to work for a variety of tasks, it extracts generalizable patterns from the demos and performs in a well-calibrated manner. This experiment demonstrates that ICL and SFT achieve adaptation through different means, and SFT should not be expected to work as a drop-in replacement for ICL. Motivated by these differences, we ask the following research question:

> *Can the information rich ICL activations be used to improve the quality of SFT?*

Next, we will show that this is indeed the case and describe our proposed SFT procedure.

## 4    INDUCING ICL-LIKE BEHAVIOR IN SFT MODELS

In this section, we will discuss our two-step SFT pipeline. First, we propose our priming step (`IA2`) which incentivizes ICL-like functional behavior. Then, we discuss how performing further SFT on these ICL-aligned weights results in much better adaptation qualities.

**`IA2`—ICL Activation Alignment:** In §3, we found SFT to have little `asim` with ICL. To increase `asim`, and hence the functional similarity with ICL, we design the following goal (`IA2`) for each layer of $M_\Theta$:

Find $\tilde{W}_{QKVO} = (\tilde{W}_Q, \tilde{W}_K, \tilde{W}_V, \tilde{W}_O)$ such that for every newly generated token (at index: $-1$):

$$\text{SA}([I \circ X]; W_{QKVO}) \approx \text{SA}(X; \tilde{W}_{QKVO}) \quad \forall X \in \mathcal{T}. \tag{3}$$

Prior work (Chen et al., 2024a) has shown a closed-form solution for the above on linearized attention models (using a kernel approximation for softmax). We aim to find a practical general solution in real non-linear transformers. Our goal tries to achieve the qualities of ICL directly in the weights of the SFT model. These modified weights are not incentivized to produce similar outputs, but to process all inputs (queries) similar to how ICL does at each layer of the base model.

To perform `IA2`, we first generate model responses using ICL ($T_i = [I \circ X_i]$), giving us $\hat{Y}_i$ for all training samples $X_i$. We use the remaining samples in the dataset to construct $I$ for each sample. We collect activations at each output position giving us $A_{\text{ICL}}^i \in \mathbb{R}^{L \times G \times d}$ assuming $G$ response tokens. Next, we provide the response attached only with the query $T_i = [X_i \circ \hat{Y}_i]$, as if the model had produced the ICL response with only the query in context. This gives an unaligned activation tensor $A^i$ at the output token positions. Then, we simply train the model to minimize the mean squared error between the two activations w.r.t. model parameters:

$$\mathcal{L}_{\texttt{IA2}} = \sum_{i=1}^{N} ||A^i - A_{\text{ICL}}^i||. \tag{4}$$

**`IA2` → SFT:** After `IA2`, we continue to train the model with the standard cross-entropy loss for SFT (Equation 2) on target tokens. Say `IA2` training updates the model parameters from $\Theta$ to $\Theta'$. We collect output responses $\bar{Y}_i$ for all $X_i$ using $M_{\Theta'}$ and minimize the SFT loss between $\bar{Y}_i$ and ground truth $Y_i$ for all training samples. This loss further aligns the model's outputs with our intended targets. Overall, our proposed training pipeline has the following two steps:

1. Collect target ICL activations $A_{\text{ICL}}$ and perform `IA2` using $\mathcal{L}_{\texttt{IA2}}$ until convergence. This aligns the model's functional behavior with ICL.
2. Switch to $\mathcal{L}_{\text{SFT}}$ with ground truth target responses $Y$, and train until convergence to align with target output behavior.

## 5    EXPERIMENTAL RESULTS

Our study includes experiments in two different settings: single-token and multi-token, with different characteristics. We will highlight any differences as we present our results.

**Adaptation Tasks:** In the single-token setting, the output $Y_i$ for each sample is a single token, typically a category/class label, True/False, or MCQ choice. In the multi-token setting, the output $Y_i$ can be multiple tokens long and variable in length. This is the general case for open-ended generation. For both settings, we test and compare our proposed SFT pipeline on various tasks using multiple models. In many cases, we also test the adaptation methods on OOD datasets to study their behavior under distribution shift. We enlist the tasks under each category below:

**Single-token tasks:**

- *Sentiment Classification:* We use SST2 (Socher et al., 2013), Financial Phrasebank [FinS] (Malo et al., 2014) and Poem Sentiment [PoemS] (Sheng & Uthus, 2020) datasets. We use SST2 and FinS for training models and use all three for evaluation on each set. This creates one in-distribution (ID) and 2 out-of-distribution (OOD) evaluation datasets.
- *True/False:* We use StrategyQA [STF] (Geva et al., 2021) which involves multi-hop reasoning before coming to a True/False conclusion.
- *News Categorization:* We use AGNews [AGN] (Zhang et al., 2015) and BBCNews [BBCN] (Greene & Cunningham, 2006). We train on AGN and evaluate on both.
- *MCQ:* We use SciQ (Johannes Welbl, 2017), and QASC (Khot et al., 2019). To create additional complexity, we remap the MCQ choices [A, B, C ...] to an unrelated token set. Hence, we call the datasets SciQr and QASCr. We train on SciQr and evaluate on both.

**Multi-token tasks:**

- *Grade-school Math:* We use GSM8K (Cobbe et al., 2021), and GSM Symbolic [GSM8Ks] (Mirzadeh et al., 2024). We train on GSM8K and evaluated on both.
- *Advanced Math:* We use MATH Algebra [HMathA] (Hendrycks et al., 2021) which consists of higher grade Algebra problems.
- *Scientific QA:* We used SciQ again, but this time for generating the tokens in the the answer instead of choosing between options.

**Data Setup:** As our focus is on the data-scarce few-shot setting where ICL is typically used, we create multiple training datasets $\mathcal{D}_\mathcal{T} = \{(X_i, Y_i)\}_{i=1}^N$ for each task by varying $N$ in orders of $2 : [2, 4, 8, 16, ...]$. We create 5 different sets at each $N$ value to average out outlier effects on our final quantitative analysis. Importantly, *we collect ICL activation tensors $A_{ICL}^i \, \forall X_i$ for **IA2** training by using the remaining $N - 1$ samples as ICL demonstrations (in random order), effectively reusing training samples*. This makes sure that **we use the exact same data for all adaptation methods**, keeping the comparison fair. For evaluation, we sampled a different set of 500 samples for each task. Additional details about the datasets and how they are processed can be found in §A.

**Multi-token details:** Here, the model outputs a sequence of *reasoning tokens* (not inside the <think> tags, just standard text) before the answer in a specific format illustrated in the demonstrations. The demonstrations and expected output are both chain-of-thought (Wei et al., 2022) style. The actual answer needs to be extracted from the generated text to evaluate model performance and different datasets have different method for parsing the answer from response tokens.

- For GSM8K and GSM8Ks, the ground truth answers have the following pattern: "<reasoning steps> #### <numerical answer>". We do not evaluate the correctness or conciseness of the reasoning steps and only parse the numerical answer out from the response and match it exactly with the ground truth answer for calculating accuracy. We also use other common answer parsing techniques like matching with "The answer is <value>", to allow for slight variations in output format. Our exact parsing function can be seen in our code.
- For HMathA, the answers are expected to be inside a \boxed{} element and parsed accordingly.
- For SciQ, we generate the ground truth format similar to the GSM8K pattern, i.e., "<support text> #### <text answer>". We parsed the answers according to this format and considered an exact string match as the only correct answer.

**Training Setup:** We use two model families to test each training method across all tasks. We use the Qwen3-4B-Base model on every task (single/multi-token). In addition, we also use the Llama-3.2-1B model on every single-token task except SciQr for which we use Llama-3.2-3B. This is because the 1B Llama model was unable to improve above random baseline for any $N$ upto 128. For the multi-token tasks, we use the Llama-3.2-1B-Instruct model as the secondary model. Using

| Dataset | | Adaptation Method | | | | | | | |
| --- | --- | --- | --- | --- | --- | --- | --- | --- | --- |
| | | ICL | | SFT only | | **IA2** only | | **IA2** → SFT | |
| Source | Eval | acc ↑ | ece ↓ | acc ↑ | ece ↓ | acc ↑ | ece ↓ | acc ↑ | ece ↓ |
| AGN | AGN | 30.0 (02.3) | 0.13 (0.03) | 27.9 (04.3) | 0.52 (0.12) | 24.0 (00.3) | **0.11** (0.01) | **31.8** (03.8) | 0.35 (0.24) |
| | BBCN* | 28.8 (00.6) | 0.12 (0.01) | **31.8** (07.1) | 0.56 (0.14) | 24.3 (01.3) | **0.10** (0.02) | 27.9 (04.3) | 0.40 (0.21) |
| FinS | FinS | 63.6 (01.5) | 0.12 (0.01) | 67.4 (14.1) | 0.31 (0.14) | 63.1 (15.0) | 0.24 (0.06) | **78.7** (13.7) | **0.16** (0.11) |
| | PoemS* | 56.9 (02.2) | 0.12 (0.02) | 49.9 (04.6) | 0.48 (0.04) | 52.8 (05.6) | **0.15** (0.04) | 60.6 (15.7) | 0.36 (0.15) |
| | SST2* | 70.1 (01.3) | 0.17 (0.01) | 59.4 (06.5) | 0.20 (0.10) | 69.8 (13.7) | 0.30 (0.10) | **71.4** (18.9) | **0.20** (0.15) |
| SciQr | QASCr* | 56.5 (01.4) | 0.08 (0.01) | 76.5 (03.8) | 0.09 (0.05) | 71.2 (03.0) | 0.12 (0.01) | **79.4** (01.8) | **0.09** (0.01) |
| | SciQr | 87.7 (01.1) | 0.07 (0.01) | 88.3 (04.6) | 0.07 (0.05) | 90.4 (01.5) | 0.09 (0.01) | **91.7** (01.7) | **0.05** (0.01) |
| SST2 | FinS* | 41.9 (00.3) | 0.19 (0.01) | 68.4 (02.3) | 0.30 (0.04) | 71.3 (02.1) | 0.21 (0.03) | **82.4** (12.2) | **0.11** (0.08) |
| | PoemS* | 65.1 (01.2) | 0.11 (0.02) | 56.5 (13.2) | 0.33 (0.14) | 62.4 (09.2) | **0.19** (0.06) | **68.4** (08.4) | 0.30 (0.08) |
| | SST2 | 85.4 (00.4) | 0.13 (0.00) | 65.2 (10.7) | 0.22 (0.10) | 82.7 (06.4) | 0.28 (0.03) | **90.4** (02.3) | **0.06** (0.01) |
| STF | STF | 66.0 (02.1) | 0.07 (0.01) | 52.4 (04.4) | 0.29 (0.10) | 62.2 (04.1) | **0.16** (0.04) | 62.4 (04.3) | 0.29 (0.08) |

Table 1: Performance report for $N = 4$ on Qwen3-4B-Base model, showing accuracy (acc) and Expected Calibration Error (ece). Numbers in parentheses show standard deviations across 5 runs for the best performing learning rate. Best training method shown in **bold**. (*) highlights OOD evaluations. *Our proposed **IA2** → SFT training method outperforms standard SFT across the board.*

an Instruct model tests our method's efficacy on post-trained models. For training models, we use LoRAs with rank 8 to modify the $W_Q, W_K$ and $W_O$ matrices of self-attention layers. We chose these models and weight combinations reasonably on the basis of available resources, as we trained more than 13,000 models in total for our experiments. Training in multi-token setting is much more resource extensive. As each sample has a long answer, we decided to only use $N$ upto 16 for Llama and $N$ upto 8 for Qwen models in multi-token setting. We also fix the maximum generated tokens ($G = 200$) during training for efficiency. This is used in **IA2** for collecting the activation tensor $\hat{A}_i \in \mathbb{R}^{L \times 200 \times d} \ \forall x_i$. For SFT, $G$ is equal to the number of tokens in the ground truth answer. Each (method, dataset) combination is trained until convergence (upto 50 epochs) with 3 learning rates (slow:`1e-4`, medium:`3e-4`, fast:`1e-3`), and the best performing one is chosen to neutralize the effect of hyperparameter selection on method performance. Additional details in §B.

**Evaluation:** In the single-token setting, we use accuracy and expected calibration error (ECE) as performance metrics. Given that the model's answer is a single token, we calculate the accuracy and ECE on the basis of the first generated token's probability distribution. First, we evaluate the base model with ICL demos to set the ICL performance benchmark. For this, we evaluate the validation set of each task 5 times with different ICL demos to report their average metrics. Then, we evaluate each training method without ICL demos and report their best-performing average metrics. In the multi-token setting, we restricted the maximum generated tokens during evaluation to judiciously use compute resources. We set the limit to 200 for GSM8K and GSM8Ks datasets. This is on the basis of their 95 percentile answer lengths (184 and 154 respectively). We relax this limit to 400 for HMathA, as the 95 percentile answer lengths are 307. For performance, we consider only accuracy, i.e., number of questions answered correctly in the required format (see answer parsing and other details in §A). As the answer confidence is hard to measure for open-ended generation tasks, we do not measure calibration.

## 5.1 **IA2** PRIMING BOOSTS SFT PERFORMANCE

**Single-Token results:** In Table 1, we report the performance metrics of adapted Qwen models for the practical setting of $N = 4$ for single-token tasks. We find that *our proposed **IA2** → SFT method outperforms SFT only training* in almost all cases in terms of *both accuracy and calibration*, all while using the same amount of data. In most cases, **IA2** → SFT also outperforms ICL in terms of accuracy, with slightly less calibration. Notice that **IA2** only training reaches high accuracy in many cases, while being reasonably calibrated like ICL, *without being trained on target response tokens*. This highlights the richness of **IA2** training signal. While this table shows the results for one setup, Figure 1 captures the aggregate performance trends across models and datasets. Detailed tabulated results for some other model/$N$ combinations, and a statistical significance test of our improvement can be seen in §C.

| Dataset | | Adaptation Method | | | | |
|---------|------|------------------|-----------|-----------|----------|--------------------|
| Source | Eval | w/o ICL acc ↑ | ICL acc ↑ | SFT only acc ↑ | **IA2** only acc ↑ | **IA2** → SFT acc ↑ |
| GSM8K | GSM8K | 56.4 (00.0) | 76.4 (01.1) | 70.9 (02.8) | **77.4** (02.8) | 73.6 (03.7) |
| | GSM8Ks* | 45.4 (00.0) | 68.4 (01.1) | 64.5 (04.0) | 66.2 (03.7) | **68.8** (05.1) |
| HMathA | HMathA | 21.0 (00.0) | 60.4 (02.1) | 50.4 (01.8) | 47.8 (06.0) | **55.3** (02.1) |
| SciQ | SciQ | 15.4 (00.0) | 37.5 (01.0) | 35.0 (07.1) | 06.9 (13.8) | **40.8** (07.6) |

Table 2: Performance report for $N = 4$ on Qwen3-4B-Base models, on multi-token datasets. Best training method shown in **bold**. *IA2 → SFT outperforms standard SFT across all datasets.*

**Multi-Token results:** In Table 2, we report the accuracy of adapted Qwen models in the multi-token setting. Like single-token, our proposed *IA2 → SFT method outperforms SFT only training, in all cases*. It is noteworthy that compared to the single-token setting, **IA2** → SFT performs worse than ICL sometimes. We suspect that this is due to imperfect compression of longer contexts in the same small LoRA weight space, which should improve with larger ranks. More tabulated results for other model/$N$ combinations are present in §C.

## 5.2 WHY IS **IA2** PRIMING IMPORTANT?

We investigate several properties of the three training methods: 1) SFT only, 2) **IA2** only and 3) **IA2** → SFT models, and make the following observations.

**Activation Similarity vs Performance:** Figure 2 shows the activation similarity of all 3 methods with ICL. **IA2** drastically increases this similarity as expected. Importantly, **IA2** → SFT retains higher similarity with ICL activations even after SFT training. In Figure 3 we show the relationship between ICL activation similarity and performance metrics. **IA2** → SFT sits comfortably between **IA2** and SFT in terms of activation similarity, while achieving better accuracy and calibration. Importantly, note that SFT only and **IA2** → SFT training only differs in the start state. This implies that **IA2** *offers a rich training signal which is unavailable via SFT only training*. It is clear that more activation similarity implies better calibration. However, extreme activation similarity may not be the optimal for achieving the best accuracy, as the ICL signal is not always right. In fact, in §C, we show one case where **IA2** models follow ICL performance even if becomes worse due to overfitting. Therefore, it is best to combine the two training signals: **IA2** to improve the internal functional alignment with ICL and SFT on ground truth responses to align with expected output behavior.

**Subspace Overlap:** Although activation patterns are a footprint of the model's functional behavior, it does not isolate changes in behavior induced by different training methods. This is because we use LoRA adaptors and the activation patterns have significant influence from the unchanged base model weights. To isolate the change in behavior, we analyze LoRA weights directly.

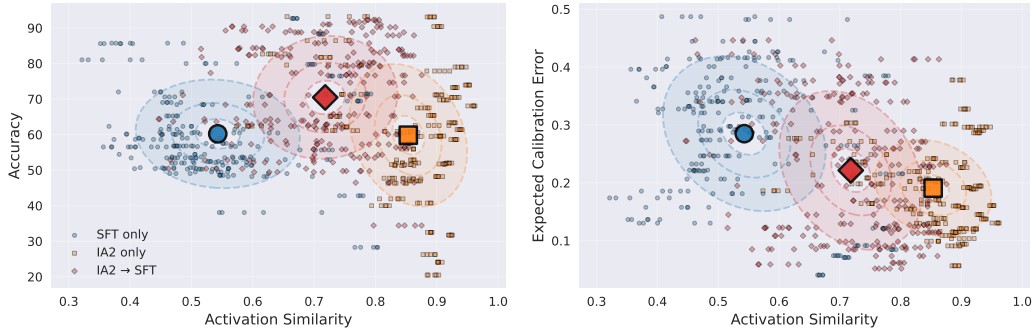

Figure 3: We scatter $\mathtt{asim}\,(A, A_{\mathrm{ICL}})$ vs Accuracy/ECE for all training methods to show the impact of $\mathtt{asim}$ on performance metrics. Each point corresponds to one training experiment. With increasing similarity, ECE goes down smoothly. But extreme ICL activation alignment may leave some accuracy gains on the table that can be sourced using SFT on ground truth responses.

Figure 4 shows the weight subspace overlap distribution between pairs of models trained using different methods. We use models from all single/multi token experiments. We first perform SVD on the weight matrices to find basis vectors. Then we calculate subspace overlap by the formula: $\frac{1}{r} * \|U_1^T \cdot U_2\|_F^2$ where $U_1$ and $U_2$ are the basis vectors corresponding to the two methods being compared, and $r$ is the rank of LoRA weights. Subspace overlap measures how much of the weight space is shared between two methods, in effect isolating and comparing the difference in their functional behavior. We find that *SFT only training results in weight updates which are almost completely orthogonal to the other two methods.* Meanwhile, the most performant `IA2` → SFT models share around $39\%$ of their spanned weight space on average with `IA2` only trained models. This implies that a lot of the performance gains of `IA2` → SFT models are achieved because of `IA2`, and the subspaces identified by `IA2` are practically unreachable by SFT only training. The high spread of subspace overlap between `IA2` only and `IA2` → SFT models is because in many cases, the `IA2` model itself was already output aligned, so the secondary SFT training did not change the model much before convergence, resulting in very high subspace overlap.

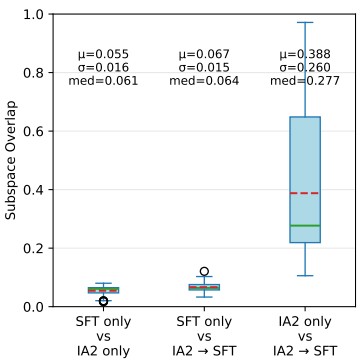

Figure 4: `IA2` → SFT models have significantly high subspace overlap with `IA2` only models, while SFT only models do not align with either. This indicates a training signal in `IA2` that is absent from SFT only training and, which is important for performance.

## 6 DISCUSSION

**`IA2` + SFT:** One natural alternative to `IA2` → SFT training is `IA2` + SFT, i.e., using both $\mathcal{L}_{\texttt{IA2}}$ and $\mathcal{L}_{\text{SFT}}$ at the same time rather than sequentially. This approach poses a practical challenge. $\hat{A}$ collected as the target activation tensor for `IA2` uses the model's own generated response tokens for any $G > 1$. We call it the ICL response, which could be very different (even in length) from the ground truth response in the dataset. This makes the two objectives incompatible to train together. However, we could perform self-distillation using ICL responses, similar to Snell et al. (2022) but including an additional `IA2` signal. We perform this training with a unified loss $\mathcal{L}_{\texttt{IA2}} + \beta \cdot \mathcal{L}_{\text{SFT}}$ and vary $\beta$ with 4 values spanning the spectrum between `IA2` only and SFT only training smoothly (details in §B). After choosing the best performing (learning rate, $\beta$) combination across 5 random seeds, we report the performance for multi-token experiments in Table 3. Note that the performance of SFT only and `IA2` → SFT training drops when compared to training on ground truth responses (from Table 2), but `IA2` + SFT extracts significantly more performance out of ICL responses when compared to SFT only training. This highlights the richness of the `IA2` training signal. Additional results of `IA2` + SFT training can be found in §C.

| Dataset | | Adaptation Method | | | | | |
|---|---|---|---|---|---|---|---|
| Source | Eval | w/o ICL acc ↑ | ICL acc ↑ | SFT only acc ↑ | **`IA2` only** acc ↑ | **`IA2` → SFT** acc ↑ | **`IA2` + SFT** acc ↑ |
| GSM8K | GSM8K | 56.4 (00.0) | 76.4 (01.1) | 66.4 (18.1) | **77.4** (02.8) | 67.2 (22.1) | 77.0 (07.8) |
| | GSM8Ks* | 45.4 (00.0) | 68.4 (01.1) | 59.8 (20.2) | 66.2 (03.7) | 61.6 (24.0) | **69.0** (09.6) |
| HMathA | HMathA | 21.0 (00.0) | 60.4 (02.1) | 49.0 (03.0) | 47.8 (06.0) | **54.0** (01.7) | 53.6 (02.7) |
| SciQ | SciQ | 15.4 (00.0) | 37.5 (01.0) | 27.2 (12.0) | 06.9 (13.8) | 32.8 (10.0) | **34.5** (10.6) |

Table 3: Performance report for $N = 4$ on Qwen3-4B-Base models when trained with ICL responses instead of ground truth responses. *`IA2` + SFT performs significantly better than SFT only highlighting the significance of `IA2`.*

**Knowledge Distillation baseline:** A popular approach to distill the behavior of a stronger teacher into the student model is through soft label matching (Hinton et al., 2015). The teacher model provides a denser training signal through the probability mass associated with each token. In our work, we treat the same model (enhanced with ICL context) as the teacher. To test how `IA2` based training performs in comparison to soft label matching, we train and evaluate Llama models on

GSM8K (1B-Instruct) and SST2 (1B) with soft label matching on ICL responses. In Table 4, we observe that soft-label matching performs significantly better than standard SFT in the multi-token case almost reaching **IA2** + SFT performance, but lacks in the single-token case. This highlights the consistency of **IA2** based training in achieving high performance.

| Dataset | $N$ | ICL | | SFT only | | SFT (soft labels) | | **IA2** only | | **IA2** $\rightarrow$ SFT | | **IA2** + SFT | |
|---|---|---|---|---|---|---|---|---|---|---|---|---|---|
| | | acc $\uparrow$ | ece $\downarrow$ | acc $\uparrow$ | ece $\downarrow$ | acc $\uparrow$ | ece $\downarrow$ | acc $\uparrow$ | ece $\downarrow$ | acc $\uparrow$ | ece $\downarrow$ | acc $\uparrow$ | ece $\downarrow$ |
| GSM8K | 4 | 34.6 (00.9) | – | 28.2 (04.5) | – | 36.1 (03.5) | – | 29.3 (05.0) | – | 30.3 (03.2) | – | **37.0** (01.9) | – |
| SST2 | 8 | 77.0 (02.2) | 0.10 (0.02) | 52.3 (04.3) | 0.38 (0.14) | 55.9 (08.8) | **0.16** (0.06) | 64.9 (15.3) | 0.30 (0.05) | 64.7 (19.3) | 0.33 (0.19) | **66.9** (18.4) | 0.28 (0.21) |

Table 4: Comparison with Knowledge Distillation (KD) baseline (SFT on soft labels). Although KD baseline performs better than standard SFT in the multi-token case, **IA2** based training provides consistently high performance for single/multi token cases.

**Why IA2 based training does not consistently beat ICL?** In some multi-token cases, like the Qwen model on math datasets (GSM8K and HMathA), we find that ICL performance exceeds all training methods including **IA2** based training. We suspect that this is due to mid-training of Qwen on STEM data, making ICL extremely sample efficient (surprisingly, ICL performance for GSM8K at $N = 2$ is larger than that at $N = 4, 8$). In Table 5, we illustrate the evolution of performance with shot count. We also highlight the gap between ICL and **IA2** $\rightarrow$ SFT performance, which closes quickly with more shots. Our key takeaway—**IA2** $\rightarrow$ SFT performs better than SFT only—remains consistent as highlighted by the last column.

| Dataset | $N$ | ICL | SFT only | **IA2** only | **IA2** $\rightarrow$ SFT | Gap wrt ICL | Gap wrt SFT only |
|---|---|---|---|---|---|---|---|
| GSM8K | 2 | 81.2 (00.4) | 65.2 (01.0) | 70.7 (01.8) | 74.6 (02.4) | -06.6 | +09.4 |
| | 4 | 76.4 (01.1) | 70.9 (02.8) | 77.4 (02.8) | 73.6 (03.7) | -02.8 | +02.7 |
| | 8 | 77.2 (00.9) | 73.5 (01.6) | 78.8 (02.5) | 76.0 (02.0) | -01.2 | +02.5 |
| HMathA | 2 | 58.6 (02.0) | 47.8 (04.0) | 33.9 (06.4) | 48.7 (07.0) | -09.9 | +00.9 |
| | 4 | 60.4 (02.1) | 50.4 (01.8) | 47.8 (06.0) | 55.3 (02.1) | -05.1 | +04.9 |
| | 8 | 62.1 (01.0) | 52.0 (00.8) | 57.3 (03.7) | 59.4 (00.8) | -02.7 | +07.4 |

Table 5: ICL is more sample efficient than training methods for Qwen models in math datasets. But **IA2** $\rightarrow$ SFT quickly closes the gap in performance with increasing number of shots, and consistently beats SFT only training.

**Other fine-tuning methods:** LoRA is not the only method to fine-tune models. One could perform full fine tuning (full-rank) or other parameter efficient methods like prompt/prefix tuning (Lester et al., 2021; Li & Liang, 2021) and activation scaling ((IA)[3] Liu et al. (2022)). Our proposed pipeline is not in competition, rather an improvement over these methods as it provides better training signals. To establish this, we repeat our experiments on a few datasets with the popular PEFT method (IA)[3]. We followed the exact same methodology from §5, just replaced LoRA training with (IA)[3] (details about setup in §B). We report the metrics for one setting in Table 6 which shows trends similar to LoRA. Other results can be seen in §C. We defer the exploration of improvement in other SFT methods through **IA2** for future work. On a side note, **IA2** also marks a major improvement in method naming compared to (IA)[3].

| Dataset | | Adaptation Method | | | | | | | |
|---|---|---|---|---|---|---|---|---|---|
| Source | Eval | ICL | | SFT only | | **IA2** only | | **IA2** $\rightarrow$ SFT | |
| | | acc $\uparrow$ | ece $\downarrow$ | acc $\uparrow$ | ece $\downarrow$ | acc $\uparrow$ | ece $\downarrow$ | acc $\uparrow$ | ece $\downarrow$ |
| SciQr | QASCr* | 35.6 (01.4) | 0.06 (0.01) | 32.8 (03.4) | 0.20 (0.08) | 34.2 (02.7) | **0.08** (0.01) | **48.6** (05.5) | 0.25 (0.11) |
| | SciQr | 60.0 (01.8) | 0.07 (0.01) | 64.5 (05.0) | 0.12 (0.05) | 63.0 (04.6) | 0.10 (0.04) | **70.6** (04.6) | **0.08** (0.02) |
| SST2 | FinS* | 56.6 (01.6) | 0.17 (0.01) | 58.2 (10.7) | 0.25 (0.03) | **61.9** (09.5) | **0.18** (0.06) | 59.8 (07.8) | 0.18 (0.03) |
| | PoemS* | 52.3 (03.0) | 0.19 (0.02) | 48.1 (03.2) | 0.44 (0.11) | 48.1 (03.4) | **0.11** (0.04) | **59.8** (13.4) | 0.33 (0.13) |
| | SST2 | 60.3 (02.8) | 0.19 (0.02) | 52.4 (02.8) | 0.35 (0.05) | 54.2 (04.2) | **0.15** (0.06) | **62.6** (09.1) | 0.28 (0.11) |

Table 6: Performance report for $N = 4$ on Llama-3.2 models trained using (IA)[3]. **IA2** $\rightarrow$ SFT achieves significant gains over SFT only training, exceeding even ICL accuracy in all cases.

**Computational Overhead:** `IA2` $\rightarrow$ SFT requires a data (activations) collection step before training, where we perform ICL inference on the model with the given dataset. This is a one-time cost to collect the rich ICL activations, which greatly benefits the model's capabilities during inference that runs at the same cost as SFT only models.

**Potential improvements:** We made several design choices, which were fixed due to resource constraints, that have potential for exploration and improvements. The most important ones include ablations around LoRA parameters (rank, target modules, etc.) and selective `IA2`. We showed that extreme alignment can hurt performance and previous work (Todd et al., 2023) has shown that some layers are more sensitive to ICL than others. Therefore, selectively aligning layers can be more effective. Another minor improvement could include prompt optimization before collecting target activations, as ICL performance is sensitive to the order of demos (Lu et al., 2022; Zhao et al., 2021). Lastly, we perform the `IA2` $\rightarrow$ SFT pipeline with fixed learning rates, but the two signals could benefit from finer grained control. We defer these explorations to future work.

## 7 RELATED WORK

**ICL vs SFT:** Prior works have studied the performance differences between ICL and SFT (Mosbach et al., 2023; Duan et al., 2024), but the verdict is unclear. In our work, we found SFT to perform worse than ICL. Other works have studied the functional similarity (Dai et al., 2022) and differences (Wang et al., 2023) of ICL with SFT. Some works show how adaptation through ICL/SFT can be miscalibrated and how to improve it (Zhang et al., 2024; Li et al., 2025; 2024; Xiao et al., 2025). Sia et al. (2024) identify "where" translation happens in LLMs during ICL through context masking, using activations as a signal for functional behavior similar to our work. Importantly, recent work (Chu et al., 2025; Shenfeld et al., 2025) has shown SFT to be prone to memorization/overfitting and catastrophic forgetting. Our improved SFT pipeline aims to address these problems.

**Context Distillation:** A large body of work targets to improve LLM adaptation using some form of compression of explicit context signals like us. These range from using intermediate *contemplation* tokens for compression (Cheng & Van Durme, 2024), internalizing context through generating adapters on the fly (Chen et al., 2025; Charakorn et al., 2025), using related text to enhance distillation (Zhu et al., 2025; Choi et al., 2025), or compressing context directly into weights (Shen et al., 2025; Deng et al., 2024; Yu et al., 2024; Snell et al., 2022; Chen et al., 2024b; Shin et al., 2024). These works aim to distill the effect of given contexts on model outputs using token response signals. Our work is complementary to these works and improves the distillation effects through the model's own ICL processing signals, an idea with supporting prior work (Aguilar et al., 2020; Jin et al., 2024; Yang et al., 2024).

**Activation Steering:** Lastly, a growing line of work attempts to extract "steering" vectors or weights to apply on targeted locations inside the model to create an intended effect on the model output (Postmus & Abreu, 2024; Stolfo et al., 2025; Caccia et al., 2025; Fleshman & Van Durme, 2024; Arditi et al., 2024). These methods are similar to our work in that they intervene in the activation space to influence the model's functionality. In contrast to their targeted approach and small application domain, our method aims to incite a large behavioral change in the model which avoids potential brittleness and reliability issues (Queiroz Da Silva et al., 2025).

## 8 CONCLUSION AND FUTURE WORK

In this work, we used model activations to probe the functional behavior of LLMs and found a key distinction: ICL and SFT achieve adaptation through different mechanisms, and ICL often encodes richer, more generalizable patterns. Motivated by this, we introduced `IA2`, an SFT priming technique that enables ICL-like internal behavior in the model. This simple step shifts models into a more adaptable weight subspace, which is inaccessible to SFT alone. Extensive experiments show that `IA2` consistently improves the accuracy and calibration of SFT. In the future, we aim to test other effects like impact on diversity and catastrophic forgetting of `IA2` based SFT to measure how `IA2` can help in post-training LLMs. We also aim to study the effect of `IA2` on post-trained (RL-tuned) models.

## REPRODUCIBILITY STATEMENT

We provide codes used for data download and processing, training, evaluation, analysis, plotting, etc. on github. All details in this paper (§5 and the appendix) along with a detailed README file should be sufficient to fully reproduce our results as we used repeatable random seeds.

## ACKNOWLEDGMENTS

This work is supported by ONR (N00014-24-1-2089), an Amazon Research Award, and a grant from Open Philanthropy. We thank Andrea Wynn for their constructive feedback on earlier versions of this document.

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

# A  ADDITIONAL DATASET DETAILS

For all tasks, we shuffled and filtered out at most 2000 samples that would be used for creating training datasets (including ICL demonstrations). For the evaluation datasets, we filtered out at most 500 different samples (all datasets had 500 samples, apart from PoemS which had only 288 samples). These evaluation samples are never seen by the model in any adaptation method. Some details about datasets are presented below.

- **SST2** contains a binary positive/negative sentiment classification of phrases from movie reviews. The prompt was structured as: "Text:$< X_i >$\n Label:". The model is supposed to output the sentiment $Y_i$ as a single token from the set $[0, 1]$. We created training datasets for $N \in \{2, 4, 8, 16, 32, 64, 128\}$. We trained with Llama-3.2-1B on all $N$ settings, and with Qwen3-4B-Base on $N \in \{2, 4, 8\}$ (as the ICL performance quickly saturates).

- **FinS** contains positive/negative/neutral sentiment for financial news sentences. It is annotated by multiple humans so we choose the subset of samples where at least 50% of annotators agreed on the classification and only included positive/negative samples to keep the label set consistent with SST2. Prompt and output structure was the same as SST2. We created training datasets for $N \in \{2, 4, 8, 16, 32, 64, 128\}$. We trained with Llama-3.2-1B on all $N$ settings, and with Qwen3-4B-Base on $N \in \{2, 4, 8, 16\}$.

- **PoemS** is a small dataset of sentiment analysis on poem verses. We filtered for positive/negative sentiments and only used this dataset for OOD evaluation. Prompt and output structure was the same as SST2.

- **STF** contains True/False questions that require logical reasoning based on common sense and general knowledge facts. The prompt was then structured as: "Question:$< X_i >$\n Answer:". The model is supposed to output T/F ($Y_i$) as a single token from the set $[0, 1]$. We created training datasets for $N \in \{2, 4, 8, 16, 32\}$. We trained only with Qwen3-4B-Base on $N \in \{2, 4, 8\}$ (as the ICL performance on Llama models did not improve much).

- **AGN** contains 4 class (World, Business, Sports, Sci/Tech) classification of News articles. Prompt structure was the same as SST2. The model is supposed to output the class $Y_i$ as a single token from the set $[0, 1, 2, 3]$. We created training datasets for $N \in \{2, 4, 8, 16, 32, 64, 128\}$. We trained with both Llama-3.2-1B and Qwen3-4B-Base on all $N$ settings (for this data, performance increase with $N$ was very slow in both models).

- **BBCN** is slightly shifted containing 5 classes. We mapped the 5 BBCN classes to AGN classes as follows: Entertainment, Politics $\rightarrow$ World; Business $\rightarrow$ Business, Sports $\rightarrow$ Sports, Tech $\rightarrow$ Sci/Tech. Prompt and output structure was the same as AGN.

- **SciQr** contains 3 distractors and 1 correct choice for Science exam questions. The choice set is hence $[0, 1, 2, 3]$. However we remapped the labels to different tokens (discussed below). The choices were randomly shuffled and combined to give a choice_string like "[label$_0$]choice$_0$\n ... [label$_3$]choice$_3$". The prompt was structured as: "Question:$< X_i >$\n Choices:$<$ choice_string $>$ Answer:". The model is supposed to output the choice $Y_i$ as a single token from the choice set. We created training datasets for $N \in \{2, 4, 8, 16, 32, 64, 128\}$. We trained with Llama-3.2-3B on all $N$ settings, and with Qwen3-4B-Base on $N \in \{2, 4, 8\}$. We chose Llama 3B model instead of 1B because the latter's ICL performance did not improve above random baseline for any $N$.

- **QASCr** contains 8 choices instead of 4. This creates a unique label shift in the data therefore we evaluated SciQr models on QASCr questions as well. Prompt and output structure was the same as SciQr.

- **GSM8K** contains grade school Math problems. Prompt structure is the same as STF. The model is supposed to output a long response ending with the proposed answer in a specific format (discussed below). We created training datasets for $N \in \{2, 4, 8, 16\}$. We trained with Llama-3.2-1B-Instruct on all $N$ settings, and with Qwen3-4B-Base on $N \in \{2, 4, 8\}$.

- **GSM8Ks** is the same quality of problems from GSM8K but with numbers and symbols replaced, to minimize influence of data contamination in the pretraining stages. The prompt and output format is the same as GSM8K.

- **HMathA** consists of more advanced Algebra questions and has the same prompt/output structure as GSM8K. We created training datasets for $N \in \{2, 4, 8\}$, and trained only Qwen3-4B-Base model on these settings (because of longer evaluations).

- **SciQ** uses the same questions from SciQr but instead of posing it as a MCQ problem, we use it as longer form QA problem. We use the support text column in the dataset as reasoning tokens before using the correct answer text as the ground truth answer. The model is still given the same prompt structure as GSM8K, and it is supposed to output the support text followed by the answer text in a specific format. We created training datasets for $N \in \{2, 4, 8, 16\}$. We trained with Llama-3.2-1B-Instruct on all $N$ settings, and with Qwen3-4B-Base on $N \in \{2, 4, 8\}$.

**OOD evaluations:** For ICL OOD evaluations, we use ICL demos from the source dataset and append the query from the OOD dataset at the end to evaluate how the model handles distribution shift in the prompt. The trained models are trained with only the source data and during evaluation, as we do not use ICL demos, so the prompt only contains the query from the OOD dataset.

**Label remapping for SciQr and QASCr:** To add additional complexity to the task and reduce any influence from pretraining memorization, we remapped the MCQ choices (A, B, C ... were mapped to 0, 1, 2 ... first) using the following remapping dictionary:

```
"remap_dict": {
    "0": "apple",
    "1": "Friday",
    "2": "banana",
    "3": "Saturday",
    "4": "Thursday",
    "5": "Sunday",
    "6": "Wednesday",
    "7": "Monday"
}
```

**Multi-Token evaluation details:** During evaluation, we used a stop string "\n\n" to stop generations early as this usually indicated the end of the answer for the given question. If we do not use this stop string, models often continue generating a new query and its answer without generating the end of text token. This trick saves significant evaluation time for our experiments. We did not use publically avaliable evaluation frameworks like lm-evaluation-harness on known benchmarks like GSM8K because we wanted more fine-grained control over which samples were used in the evaluation process. As our parsing functions remain consistent across all training/adaptation methods, the relative performance between methods still gives a good estimate of how they would perform on standard evaluations.

## B  ADDITIONAL EXPERIMENTAL DETAILS

**Training details:** We stop all training runs for convergence when the loss on a held out dev set (of size $N/2$ to reflect the data scarce setting) does not decrease for 5 steps. Note that in total, we train and evaluate 15 models for every $N$ value for reliable performance numbers.

**LoRA details:** We used LoRA rank $= 8$, $\alpha = 8$ and target modules are attention weights $W_Q, W_K$ and $W_O$ for all our experiments. We settled on these parameters on the basis of generally used values for small targeted adaptations, as additional explorations require substantial compute resources for the scale of our experiments. The target modules capture the cross-attention parts $(Q, K)$ and final projection $O$ to allow for better adaptability. We did preliminary ablations to make sure our results were not biased against SFT only (next paragraph). However, these selections are subject to improvement with more exploration.

**LoRA ablations:** As our LoRA parameters are consistent, no training method has an unfair advantage. However, some methods may benefit from particular parameter choices. Therefore, we conducted small scale ablation studies to measure the impact of LoRA parameters on performance. In Table 7, we present these results for Llama 3.2-1B models trained on the SST2 dataset (evaluated on 3 datasets: SST2, FinS, PoemS) and Qwen3-4B-base models on the GSM8K dataset (evaluated on itself). Each numerical column contains [acc-mean (acc-std), ece-mean (ece-std)] across 5 re-runs for the best performing hyperparameters (learning rate, beta). Although training only QK matrices results in overall worse performance for all methods, there is still a healthy gap between SFT only and `IA2` →SFT. Rank increase from 8 to 16 does not have a huge impact on performance,

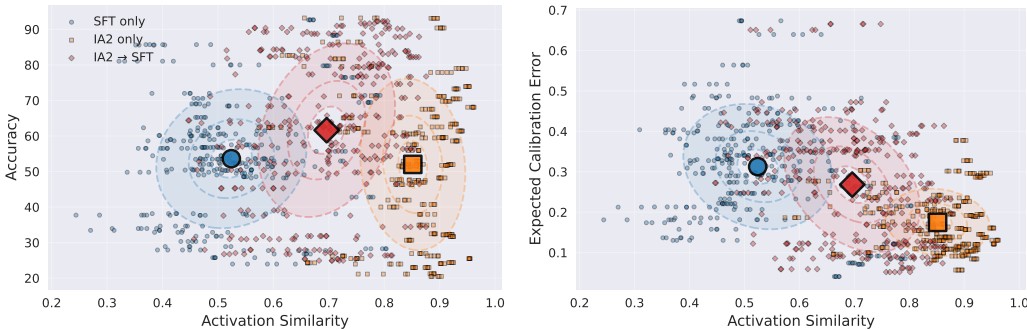

Figure 5: We scatter `asim` $(A, A_{\text{ICL}})$ vs Accuracy/ECE for all training methods to show its the impact of `asim` pn performance metrics. Each point corresponds to one training experiment. With increasing similarity, ECE goes down smoothly. But extreme ICL activation alignment may leave some accuracy gains on the table that can be sourced using SFT on ground truth responses.

and the choice between V and O matrices does not impact the results much either. Overall, our main takeaway remains consistent: **IA2** →SFT outperforms SFT only.

| Dataset | LoRA params | ICL | | SFT only | | IA2 only | | IA2 →SFT | |
|---|---|---|---|---|---|---|---|---|---|
| | | acc ↑ | ece ↓ | acc ↑ | ece ↓ | acc ↑ | ece ↓ | acc ↑ | ece ↓ |
| FinS | r8, qko | 60.2 (03.0) | 0.10 (0.02) | 60.0 (13.6) | 0.33 (0.11) | 53.2 (20.4) | 0.33 (0.04) | 89.8 (09.0) | 0.09 (0.08) |
| PoemS | r8, qko | 57.5 (01.8) | 0.11 (0.02) | 50.7 (03.7) | 0.41 (0.07) | 61.2 (12.5) | 0.22 (0.06) | 84.8 (08.1) | 0.14 (0.08) |
| SST2 | r8, qko | 77.0 (02.2) | 0.10 (0.02) | 53.5 (05.0) | 0.34 (0.11) | 64.9 (15.3) | 0.30 (0.05) | 90.4 (01.3) | 0.09 (0.02) |
| FinS | r8, qk | 60.2 (03.0) | 0.10 (0.02) | 48.8 (16.1) | 0.21 (0.12) | 51.9 (12.6) | 0.14 (0.02) | 50.4 (21.6) | 0.27 (0.07) |
| PoemS | r8, qk | 57.5 (01.8) | 0.11 (0.02) | 48.9 (03.1) | 0.24 (0.14) | 42.6 (06.2) | 0.13 (0.04) | 56.7 (12.7) | 0.29 (0.13) |
| SST2 | r8, qk | 77.0 (02.2) | 0.10 (0.02) | 50.0 (06.4) | 0.33 (0.13) | 61.6 (08.3) | 0.17 (0.09) | 65.7 (15.0) | 0.27 (0.14) |
| FinS | r8, qkv | 60.2 (03.0) | 0.10 (0.02) | 54.2 (15.8) | 0.34 (0.16) | 66.2 (01.9) | 0.16 (0.06) | 87.3 (05.4) | 0.06 (0.02) |
| PoemS | r8, qkv | 57.5 (01.8) | 0.11 (0.02) | 53.4 (05.5) | 0.36 (0.11) | 57.4 (08.0) | 0.20 (0.08) | 80.0 (07.2) | 0.19 (0.07) |
| SST2 | r8, qkv | 77.0 (02.2) | 0.10 (0.02) | 53.5 (03.4) | 0.34 (0.08) | 65.3 (18.6) | 0.34 (0.03) | 88.7 (02.8) | 0.10 (0.03) |
| FinS | r16, qko | 60.2 (03.0) | 0.10 (0.02) | 55.4 (13.0) | 0.25 (0.12) | 51.7 (19.4) | 0.33 (0.04) | 86.8 (09.4) | 0.13 (0.09) |
| PoemS | r16, qko | 57.5 (01.8) | 0.11 (0.02) | 54.5 (10.9) | 0.26 (0.16) | 57.2 (09.1) | 0.27 (0.06) | 79.5 (10.0) | 0.13 (0.05) |
| SST2 | r16, qko | 77.0 (02.2) | 0.10 (0.02) | 55.8 (13.9) | 0.29 (0.14) | 65.5 (14.9) | 0.30 (0.05) | 87.7 (05.8) | 0.12 (0.06) |
| GSM8K | r8, qko | 76.4 (01.1) | – | 70.9 (02.8) | – | 77.4 (02.8) | – | 73.6 (03.7) | – |
| GSM8K | r16, qko | 76.4 (01.1) | – | 70.3 (05.5) | – | 77.8 (06.4) | – | 74.1 (03.3) | – |

Table 7: Ablation results on LoRA parameters (rank, target modules).

**Collecting activations:** We inserted hooks in the attention modules of the models and stacked activations from multiple token positions and layers together to get the activation tensors. For the `asim` experiment, we used a small subset of validation samples (100 out of the total 500) to collect activations. Also, we restricted the trained models used in `asim` comparison to a small set corresponding to $N$ values of $\{2, 4, 8, 16\}$. This is because collecting activations with ICL on higher $N$ values takes a long time due to longer context. We used all datasets and model combinations to calculate the `asim` in Figure 2. However, we removed the multi-token experiments and AGN dataset experiments from Figure 3. This is because 1) we don't have ECE for multi-token experiments so it would make the two plots incompatible, and 2) most AGN models did not improve in performance much by $N = 16$ mark, so it introduces noisy points in the scatter plot at baseline accuract and high ECE. We show the plots including AGN in Figure 5, which still show similar trends with a few noisy points from AGN models.

**IA2 + SFT details:** The $\beta$ parameter used in **IA2** + SFT experiments depends on the magnitude of activation tensors generated by the model. We found that the Llama family had a small activation magnitude, hence the MSE loss $\mathcal{L}_{\text{IA2}}$ at the start of training is typically an order of magnitude smaller than the cross-entropy (CE) loss $\mathcal{L}_{\text{SFT}}$, therefore we choose to weight the losses accordingly and experiment with $\beta \in [0.001, 0.01, 0.05, 0.5]$. Here a $\beta = 0.5$ makes the CE loss outweigh the MSE loss almost completely. Qwen models typically have a similar order of magnitude between MSE and CE loss. Therefore, we experiment with $\beta \in [0.1, 0.5, 0.7, 0.9]$ to weight the CE loss uniformly over the spectrum. Overall, these experiments were 4 times more compute extensive because for each learning rate, we train the models with 4 different values of $\beta$. We also notice that

| | **IA2** $\to$ SFT $>$ SFT only (acc) | | **IA2** $\to$ SFT $<$ SFT only (ece) | |
|---|---|---|---|---|
| $N$ | t-statistic | p-value | t-statistic | p-value |
| **2** | 2.3293 | 0.02178 | 2.1009 | 0.03806 |
| **4** | 6.1933 | 1.348e-08 | 4.4248 | 2.484e-05 |
| **8** | 4.3758 | 2.885e-05 | 2.4725 | 0.01504 |
| **16** | 6.4200 | 9.38e-09 | 1.7433 | 0.08518 |
| **32** | 7.3954 | 3.752e-10 | 1.3014 | 0.1978 |
| **64** | 6.3315 | 3.621e-08 | 2.2109 | 0.03093 |
| **128** | 8.0807 | 3.998e-11 | 2.9682 | 0.004323 |

| | **IA2** $\to$ SFT $>$ SFT only (acc) | |
|---|---|---|
| $N$ | t-statistic | p-value |
| **2** | 6.0541 | 7.328e-07 |
| **4** | 6.6374 | 1.296e-07 |
| **8** | 4.7901 | 3.209e-05 |
| **16** | 1.0453 | 0.3136 |

Table 8: We show the statistical significance of improvement in performance of **IA2** $\to$ SFT over SFT only. The stats for single-token experiments are on the left and multi-token experiments on the right. Statistically significant ($p < 0.05$) improvements are marked with green and insignificant improvements with red.

in some cases, **IA2** + SFT or even **IA2** $\to$ SFT models trained with ICL response tokens outperform those trained with ground truth tokens (see tables in §C). We suspect that this may be due to ICL response being more "natural" (with respect to pre-trained weights) to the model and easily aligned compared to potentially surprising ground truth responses.

**(IA)$^3$ details:** We used the same target modules, $W_Q, W_K$ and $W_O$ as LoRA and used a different learning rate range $[0.1, 0.01, 0.001]$. This is because we experimented and found that (IA)$^3$ typically needs a faster learning rate than LoRA to work. (IA)$^3$ works differently from LoRA – it does not modify model weights, rather introduces attention vectors to scale the output of the target modules. But our training procedure works seamlessly with this as well, as it finds better scaling vectors through the rich **IA2** training signal.

**Performance numbers in Figure 1:** To create the bar plots in figure 1, we only considered single-token experiments (because of ECE) and included results from both Qwen/Llama models across datasets. The performance numbers for different datasets have signficant variations at a single $N$ value but we show the average across all datasets. To make sure that these results are statistically significant, we also performed paired t-tests between SFT only and **IA2** $\to$ SFT performance metrics across all datasets and training runs. We report this statistical significance result in Table 8, which shows that almost all cases of improvements were statistically significant.

**ICL overfitting:** ICL is not perfect and is prone to overfitting. We see this in our OOD label shift scenario QASCr evaluation based on SCiQr data. In Figure 6, we show the trend of performance of all adaptation methods over variation in dataset size $N$, and notice the impact of ICL overfitting.

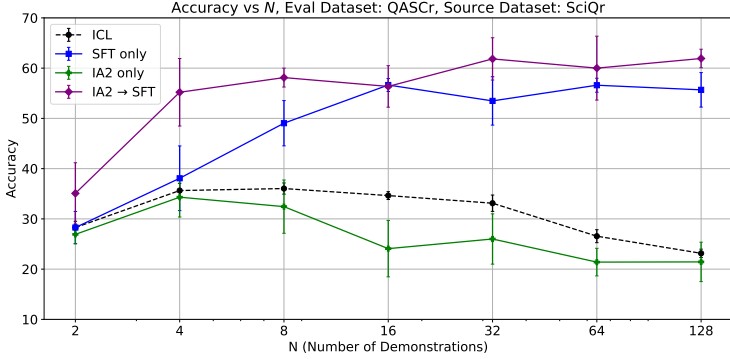

Figure 6: The performance of ICL drops with increasing $N$ when the query is OOD to the demonstration data. **IA2** only performance follows the ICL curve indicating a strong correlation in its functional behavior with ICL. However, this does not impact **IA2** $\to$ SFT performance.

## C  ADDITIONAL RESULTS

We show more performance tables here for the reader's perusal.

| Dataset | | Adaptation Method | | | | | | | | | |
|---|---|---|---|---|---|---|---|---|---|---|---|
| | | ICL | | SFT only | | **IA2** only | | **IA2** $\rightarrow$ SFT | | **IA2** + SFT | |
| Source | Eval | acc $\uparrow$ | ece $\downarrow$ | acc $\uparrow$ | ece $\downarrow$ | acc $\uparrow$ | ece $\downarrow$ | acc $\uparrow$ | ece $\downarrow$ | acc $\uparrow$ | ece $\downarrow$ |
| AGN | AGN | 30.0 (02.3) | 0.13 (0.03) | 21.3 (01.8) | 0.66 (0.06) | 24.0 (00.3) | **0.11** (0.01) | 23.9 (01.5) | 0.70 (0.08) | **24.2** (00.9) | 0.26 (0.10) |
| | BBCN* | 28.8 (00.6) | 0.12 (0.01) | 30.8 (05.6) | 0.52 (0.08) | 24.3 (01.3) | **0.10** (0.02) | 30.1 (05.3) | 0.47 (0.19) | **31.3** (05.6) | 0.39 (0.16) |
| FinS | FinS | 63.6 (01.5) | 0.12 (0.01) | 61.9 (11.4) | 0.35 (0.05) | **63.1** (15.0) | **0.24** (0.06) | 60.8 (13.5) | 0.35 (0.05) | 62.8 (09.7) | 0.31 (0.04) |
| | PoemS* | 56.9 (02.2) | 0.12 (0.02) | 47.7 (03.1) | 0.50 (0.06) | **52.8** (05.6) | **0.15** (0.04) | 47.4 (02.5) | 0.47 (0.13) | 47.8 (03.2) | 0.25 (0.08) |
| | SST2* | 70.1 (01.3) | 0.17 (0.01) | 53.0 (01.7) | 0.34 (0.11) | **69.8** (13.7) | **0.30** (0.10) | 51.3 (01.8) | 0.49 (0.02) | 54.1 (03.8) | 0.37 (0.09) |
| SciQr | QASCr* | 56.5 (01.4) | 0.08 (0.01) | 66.6 (15.1) | 0.13 (0.09) | 71.2 (03.0) | 0.12 (0.01) | **72.8** (06.6) | 0.14 (0.03) | 68.8 (02.7) | **0.08** (0.02) |
| | SciQr | 87.7 (01.1) | 0.07 (0.01) | 80.6 (14.3) | 0.12 (0.10) | **90.4** (01.5) | 0.09 (0.01) | 86.6 (07.5) | **0.08** (0.05) | 89.5 (00.8) | 0.08 (0.01) |
| SST2 | FinS* | 41.9 (00.3) | 0.19 (0.01) | 57.4 (12.6) | 0.40 (0.11) | 71.3 (02.1) | **0.21** (0.03) | 66.2 (18.1) | 0.26 (0.17) | **71.5** (22.2) | 0.26 (0.23) |
| | PoemS* | 65.1 (01.2) | 0.11 (0.02) | 53.4 (07.0) | 0.40 (0.07) | 62.4 (09.2) | **0.19** (0.06) | 63.3 (11.7) | 0.35 (0.13) | **66.1** (15.0) | 0.23 (0.16) |
| | SST2 | 85.4 (00.4) | 0.13 (0.00) | 62.3 (12.6) | 0.34 (0.13) | **82.7** (06.4) | 0.28 (0.03) | 72.1 (18.4) | 0.27 (0.19) | 76.8 (14.3) | **0.27** (0.04) |
| STF | STF | 66.0 (02.1) | 0.07 (0.01) | 53.5 (06.5) | 0.41 (0.12) | **62.2** (04.1) | **0.16** (0.04) | 60.0 (07.0) | 0.36 (0.10) | 58.5 (06.4) | 0.27 (0.13) |

Table 9:  Performance report for $N = 4$ on Qwen3-4B-Base models trained using LoRA with ICL responses on single-token datasets. With ICL responses, SFT signal from tokens is not very helpful in increasing performance of **IA2** only models.

| Dataset | | Adaptation Method | | | | |
|---|---|---|---|---|---|---|
| | | w/o ICL | ICL | SFT only | **IA2** only | **IA2** $\rightarrow$ SFT |
| Source | Eval | acc $\uparrow$ | acc $\uparrow$ | acc $\uparrow$ | acc $\uparrow$ | acc $\uparrow$ |
| GSM8K | GSM8K | 56.4 (00.0) | 81.2 (00.4) | 65.2 (01.0) | 70.7 (01.8) | **74.6** (02.4) |
| | GSM8Ks* | 45.4 (00.0) | 71.3 (01.4) | 58.6 (04.3) | 62.4 (02.5) | **67.2** (03.4) |
| HMathA | HMathA | 21.0 (00.0) | 58.6 (02.0) | 47.8 (04.0) | 33.9 (06.4) | **48.7** (07.0) |
| SciQ | SciQ | 15.4 (00.0) | 36.6 (01.6) | 36.7 (06.8) | 05.7 (09.5) | **40.4** (06.2) |

Table 10:  Performance report for $N = 2$ on Qwen3-4B-Base models trained using LoRA with ground truth tokens on multi-token datasets.

| Dataset | | Adaptation Method | | | | | |
|---|---|---|---|---|---|---|---|
| | | w/o ICL | ICL | SFT only | **IA2** only | **IA2** $\rightarrow$ SFT | **IA2** + SFT |
| Source | Eval | acc $\uparrow$ | acc $\uparrow$ | acc $\uparrow$ | acc $\uparrow$ | acc $\uparrow$ | acc $\uparrow$ |
| GSM8K | GSM8K | 56.4 (00.0) | 81.2 (00.4) | 72.8 (02.0) | 70.7 (01.8) | 74.6 (02.5) | **78.8** (02.0) |
| | GSM8Ks* | 45.4 (00.0) | 71.3 (01.4) | 68.0 (03.1) | 62.4 (02.5) | 65.4 (04.8) | **71.4** (03.8) |
| HMathA | HMathA | 21.0 (00.0) | 58.6 (02.0) | 41.6 (06.4) | 33.9 (06.4) | **45.4** (04.3) | 43.9 (03.3) |
| SciQ | SciQ | 15.4 (00.0) | 36.6 (01.6) | 26.1 (07.8) | 05.7 (09.5) | 32.9 (09.1) | **33.9** (07.4) |

Table 11:  Performance report for $N = 2$ on Qwen3-4B-Base models trained using LoRA with ICL responses on multi-token datasets.

| Dataset | | Adaptation Method | | | | | | | |
|---|---|---|---|---|---|---|---|---|---|
| | | ICL | | SFT only | | IA2 only | | IA2 → SFT | |
| Source | Eval | acc ↑ | ece ↓ | acc ↑ | ece ↓ | acc ↑ | ece ↓ | acc ↑ | ece ↓ |
| AGN | AGN | 29.1 (01.0) | 0.12 (0.01) | **38.2** (05.3) | 0.30 (0.06) | 24.4 (00.4) | **0.11** (0.04) | 35.8 (06.1) | 0.39 (0.15) |
| | BBCN* | 32.6 (03.8) | 0.11 (0.03) | **34.6** (06.2) | 0.31 (0.07) | 32.1 (07.0) | **0.11** (0.06) | 29.6 (05.1) | 0.43 (0.18) |
| FinS | FinS | 57.7 (03.1) | 0.13 (0.01) | 69.0 (01.5) | 0.22 (0.08) | 61.4 (12.4) | **0.14** (0.05) | **76.6** (11.3) | 0.17 (0.09) |
| | PoemS* | 47.9 (02.6) | 0.15 (0.03) | 52.0 (04.9) | 0.29 (0.14) | 48.2 (04.0) | **0.23** (0.11) | **54.3** (14.9) | 0.34 (0.16) |
| | SST2* | 55.7 (02.7) | 0.10 (0.01) | 56.8 (04.5) | 0.27 (0.15) | 52.0 (00.5) | **0.19** (0.07) | **65.3** (15.9) | 0.26 (0.13) |
| SciQr | QASCr* | 36.0 (01.1) | 0.08 (0.01) | 49.0 (04.5) | 0.35 (0.10) | 32.4 (05.3) | **0.10** (0.01) | **58.1** (01.9) | 0.25 (0.06) |
| | SciQr | 68.5 (01.3) | 0.07 (0.01) | 70.6 (07.8) | 0.23 (0.03) | 72.6 (04.0) | 0.15 (0.06) | **82.4** (02.1) | **0.13** (0.04) |
| SST2 | FinS* | 60.2 (03.0) | 0.10 (0.02) | 60.0 (13.6) | 0.33 (0.11) | 53.2 (20.4) | 0.33 (0.04) | **89.8** (09.0) | **0.09** (0.08) |
| | PoemS* | 57.5 (01.8) | 0.11 (0.02) | 50.7 (03.7) | 0.41 (0.07) | 61.2 (12.5) | 0.22 (0.06) | **84.8** (08.1) | **0.14** (0.08) |
| | SST2 | 77.0 (02.2) | 0.10 (0.02) | 53.5 (05.0) | 0.34 (0.11) | 64.9 (15.3) | 0.30 (0.05) | **90.4** (01.3) | **0.09** (0.02) |

Table 12: Performance report for $N = 8$ on Llama-3.2 models trained using LoRA with ground truth tokens on single-token datasets.

| Dataset | | Adaptation Method | | | | | | | | | |
|---|---|---|---|---|---|---|---|---|---|---|---|
| | | ICL | | SFT only | | IA2 only | | IA2 → SFT | | IA2 + SFT | |
| Source | Eval | acc ↑ | ece ↓ | acc ↑ | ece ↓ | acc ↑ | ece ↓ | acc ↑ | ece ↓ | acc ↑ | ece ↓ |
| AGN | AGN | 29.1 (01.0) | 0.12 (0.01) | 23.4 (02.6) | 0.65 (0.14) | 24.4 (00.4) | **0.11** (0.04) | **24.4** (00.5) | 0.57 (0.15) | 24.2 (00.4) | 0.43 (0.11) |
| | BBCN* | 32.6 (03.8) | 0.11 (0.03) | 32.4 (07.0) | 0.66 (0.06) | 32.1 (07.0) | **0.11** (0.06) | 31.9 (06.9) | 0.42 (0.20) | **32.6** (06.4) | 0.61 (0.10) |
| FinS | FinS | 57.7 (03.1) | 0.13 (0.01) | 65.2 (03.0) | 0.31 (0.02) | 61.4 (12.4) | **0.14** (0.05) | 67.2 (00.8) | 0.29 (0.04) | **67.4** (00.5) | 0.26 (0.08) |
| | PoemS* | 47.9 (02.6) | 0.15 (0.03) | 47.3 (02.2) | 0.49 (0.09) | 48.2 (04.0) | **0.23** (0.11) | **49.3** (06.2) | 0.45 (0.15) | 48.8 (05.1) | 0.36 (0.12) |
| | SST2* | 55.7 (02.7) | 0.10 (0.01) | 52.6 (01.0) | 0.42 (0.10) | 52.0 (00.5) | **0.19** (0.07) | 54.3 (04.2) | 0.42 (0.09) | **54.8** (05.2) | 0.32 (0.10) |
| SciQr | QASCr* | 36.0 (01.1) | 0.08 (0.01) | 41.5 (09.0) | 0.42 (0.09) | 32.4 (05.3) | **0.10** (0.01) | **53.4** (05.2) | 0.25 (0.08) | 42.6 (02.6) | 0.30 (0.04) |
| | SciQr | 68.5 (01.3) | 0.07 (0.01) | 60.5 (12.0) | 0.34 (0.12) | 72.6 (04.0) | **0.15** (0.06) | **73.6** (09.9) | 0.21 (0.10) | 60.1 (03.0) | 0.29 (0.04) |
| SST2 | FinS* | 60.2 (03.0) | 0.10 (0.02) | 46.7 (16.8) | 0.49 (0.20) | 53.2 (20.4) | **0.33** (0.04) | **64.4** (28.3) | 0.33 (0.27) | 60.7 (24.2) | 0.35 (0.26) |
| | PoemS* | 57.5 (01.8) | 0.11 (0.02) | 52.8 (03.4) | 0.42 (0.07) | 61.2 (12.5) | **0.22** (0.06) | **66.7** (19.2) | 0.32 (0.19) | 63.1 (15.5) | 0.29 (0.11) |
| | SST2 | 77.0 (02.2) | 0.10 (0.02) | 52.3 (04.3) | 0.38 (0.14) | 64.9 (15.3) | 0.30 (0.05) | 64.7 (19.3) | 0.33 (0.19) | **66.9** (18.4) | **0.28** (0.21) |

Table 13: Performance report for $N = 8$ on Llama-3.2 models trained using LoRA with ICL response tokens on single-token datasets.

| Dataset | | Adaptation Method | | | | |
|---|---|---|---|---|---|---|
| | | w/o ICL | ICL | SFT only | IA2 only | IA2 → SFT |
| Source | Eval | acc ↑ | acc ↑ | acc ↑ | acc ↑ | acc ↑ |
| GSM8K | GSM8K | 14.6 (00.0) | 34.6 (00.9) | 22.8 (02.8) | 29.3 (05.0) | **31.5** (01.9) |
| | GSM8Ks* | 14.8 (00.0) | 27.8 (01.0) | 20.4 (03.3) | 24.2 (01.9) | **25.1** (03.4) |
| SciQ | SciQ | 02.2 (00.0) | 10.0 (00.7) | 06.3 (04.3) | 00.0 (00.1) | **12.0** (04.7) |

Table 14: Performance report for $N = 4$ on Llama-3.2 models trained using LoRA using ground truth tokens on multi-token datasets.

| Dataset | | Adaptation Method | | | | | |
|---|---|---|---|---|---|---|---|
| | | w/o ICL | ICL | SFT only | IA2 only | IA2 → SFT | IA2 + SFT |
| Source | Eval | acc ↑ | acc ↑ | acc ↑ | acc ↑ | acc ↑ | acc ↑ |
| GSM8K | GSM8K | 14.6 (00.0) | 34.6 (00.9) | 28.2 (04.5) | 29.3 (05.0) | 30.3 (03.2) | **37.0** (01.9) |
| | GSM8Ks* | 14.8 (00.0) | 27.8 (01.0) | 23.9 (03.0) | 24.2 (01.9) | 25.5 (02.9) | **27.9** (03.1) |
| SciQ | SciQ | 02.2 (00.0) | 10.0 (00.7) | 09.5 (05.8) | 00.0 (00.1) | 13.2 (07.8) | **14.8** (08.2) |

Table 15: Performance report for $N = 4$ on Llama-3.2 models trained using LoRA with ICL response tokens on multi-token datasets.

| Dataset | | Adaptation Method | | | | | | | | | |
| Source | Eval | ICL | | SFT only | | IA2 only | | IA2 → SFT | | IA2 + SFT | |
| | | acc ↑ | ece ↓ | acc ↑ | ece ↓ | acc ↑ | ece ↓ | acc ↑ | ece ↓ | acc ↑ | ece ↓ |
| SciQr | QASCr* | 35.6 (01.4) | 0.06 (0.01) | 25.9 (03.7) | 0.23 (0.09) | 34.2 (02.7) | 0.08 (0.01) | **41.5** (03.6) | 0.28 (0.13) | 33.8 (03.0) | **0.08** (0.00) |
| | SciQr | 60.0 (01.8) | 0.07 (0.01) | 53.4 (06.6) | 0.16 (0.07) | **63.0** (04.6) | **0.10** (0.04) | 54.2 (07.4) | 0.18 (0.08) | 61.9 (04.2) | 0.11 (0.03) |
| SST2 | FinS* | 56.6 (01.6) | 0.17 (0.01) | 49.2 (15.2) | 0.32 (0.15) | **61.9** (09.5) | **0.18** (0.06) | 50.7 (12.8) | 0.30 (0.13) | 52.0 (16.4) | 0.39 (0.24) |
| | PoemS* | 52.3 (03.0) | 0.19 (0.02) | 50.1 (03.4) | 0.48 (0.04) | 48.1 (03.4) | **0.11** (0.04) | 50.8 (03.8) | 0.40 (0.14) | **50.9** (06.3) | 0.17 (0.06) |
| | SST2 | 60.3 (02.8) | 0.19 (0.02) | 50.4 (02.1) | 0.40 (0.10) | **54.2** (04.2) | **0.15** (0.06) | 50.4 (02.1) | 0.45 (0.09) | 52.0 (02.5) | 0.20 (0.09) |

Table 16: Performance report for $N = 4$ on Llama-3.2 models trained using (IA)$^3$ with ICL response tokens on single-token datasets.

| Dataset | | Adaptation Method | | | | | | | |
| Source | Eval | ICL | | SFT only | | IA2 only | | IA2 → SFT | |
| | | acc ↑ | ece ↓ | acc ↑ | ece ↓ | acc ↑ | ece ↓ | acc ↑ | ece ↓ |
| SciQr | QASCr* | 36.0 (01.1) | 0.08 (0.01) | 43.4 (06.7) | 0.31 (0.09) | 36.2 (01.9) | **0.09** (0.01) | **57.7** (03.8) | 0.17 (0.04) |
| | SciQr | 68.5 (01.3) | 0.07 (0.01) | 70.0 (02.7) | **0.09** (0.03) | 69.5 (04.3) | 0.14 (0.06) | **77.4** (07.5) | 0.17 (0.08) |
| SST2 | FinS* | 60.2 (03.0) | 0.10 (0.02) | 58.5 (11.2) | 0.26 (0.10) | 59.6 (05.2) | 0.15 (0.04) | **82.1** (07.0) | **0.10** (0.08) |
| | PoemS* | 57.5 (01.8) | 0.11 (0.02) | 51.6 (04.5) | 0.37 (0.06) | 54.7 (06.2) | 0.18 (0.04) | **81.8** (09.1) | **0.14** (0.07) |
| | SST2 | 77.0 (02.2) | 0.10 (0.02) | 53.5 (02.8) | 0.25 (0.08) | 66.5 (10.1) | 0.24 (0.06) | **81.3** (12.1) | **0.17** (0.12) |

Table 17: Performance report for $N = 8$ on Llama-3.2 models trained using (IA)$^3$ with ground truth tokens on single-token datasets.

| Dataset | | Adaptation Method | | | | | | | | | |
| Source | Eval | ICL | | SFT only | | IA2 only | | IA2 → SFT | | IA2 + SFT | |
| | | acc ↑ | ece ↓ | acc ↑ | ece ↓ | acc ↑ | ece ↓ | acc ↑ | ece ↓ | acc ↑ | ece ↓ |
| SciQr | QASCr* | 36.0 (01.1) | 0.08 (0.01) | 34.8 (09.3) | 0.37 (0.11) | 36.2 (01.9) | **0.09** (0.01) | **49.3** (08.2) | 0.39 (0.07) | 36.0 (00.9) | 0.09 (0.01) |
| | SciQr | 68.5 (01.3) | 0.07 (0.01) | 52.0 (07.8) | 0.26 (0.08) | **69.5** (04.3) | 0.14 (0.06) | 66.7 (16.1) | 0.28 (0.16) | 67.0 (02.1) | **0.10** (0.03) |
| SST2 | FinS* | 60.2 (03.0) | 0.10 (0.02) | 50.0 (15.7) | 0.38 (0.21) | **59.6** (05.2) | **0.15** (0.04) | 54.0 (19.9) | 0.38 (0.20) | 51.8 (17.9) | 0.36 (0.22) |
| | PoemS* | 57.5 (01.8) | 0.11 (0.02) | 54.1 (04.6) | 0.38 (0.11) | 54.7 (06.2) | 0.18 (0.04) | **60.6** (12.0) | 0.32 (0.17) | 57.3 (10.4) | 0.27 (0.06) |
| | SST2 | 77.0 (02.2) | 0.10 (0.02) | 55.4 (07.6) | 0.36 (0.15) | **66.5** (10.1) | 0.24 (0.06) | 64.5 (18.7) | 0.31 (0.21) | 63.6 (17.3) | 0.26 (0.06) |

Table 18: Performance report for $N = 8$ on Llama-3.2 models trained using (IA)$^3$ with ICL response tokens on single-token datasets.

| Dataset | | Adaptation Method | | | | |
| Source | Eval | w/o ICL | ICL | SFT only | IA2 only | IA2 → SFT |
| | | acc ↑ | acc ↑ | acc ↑ | acc ↑ | acc ↑ |
| GSM8K | GSM8K | 14.6 (00.0) | 35.9 (01.1) | 24.6 (02.2) | 30.9 (02.3) | **31.0** (02.5) |
| | GSM8Ks* | 14.8 (00.0) | 30.1 (00.7) | 19.1 (02.0) | 23.5 (01.2) | **24.1** (02.0) |

Table 19: Performance report for $N = 8$ on Llama-3.2 models trained using (IA)$^3$ with ground truth tokens on multi-token datasets.

| Dataset | | Adaptation Method | | | | | |
| Source | Eval | w/o ICL | ICL | SFT only | IA2 only | IA2 → SFT | IA2 + SFT |
| | | acc ↑ | acc ↑ | acc ↑ | acc ↑ | acc ↑ | acc ↑ |
| GSM8K | GSM8K | 14.6 (00.0) | 35.9 (01.1) | 26.6 (10.1) | 30.9 (02.3) | 34.2 (01.3) | **34.3** (02.7) |
| | GSM8Ks* | 14.8 (00.0) | 30.1 (00.7) | 21.4 (06.6) | 23.5 (01.2) | 27.4 (01.8) | **27.4** (02.2) |

Table 20: Performance report for $N = 8$ on Llama-3.2 models trained using (IA)$^3$ with ICL response tokens on multi-token datasets.

