# OpenReview forum: "IA2: Alignment with ICL Activations improves Supervised Fine-Tuning"
_ICLR.cc/2026/Conference — ICLR 2026 Poster_

### Official Review · Reviewer_gzQ9 · 2025-10-15

**Soundness:** 3
**Presentation:** 3
**Contribution:** 3
**Rating:** 6
**Confidence:** 4

**Summary:**

The paper introduces a novel method called IA2 (Instruction Alignment with In-Context Learning Activations) to improve the performance of supervised fine-tuning for large language models (LLMs). The key idea is to leverage the activations from in-context learning (ICL) to guide the supervised fine-tuning process, thereby enhancing the model's ability to follow instructions and generalize to unseen tasks.

**Strengths:**

Novel Conceptual Framework: The paper introduces a novel idea of aligning in-context learning (ICL) activations with supervised fine-tuning. The concept of using ICL activations as a regularization mechanism during fine-tuning is a creative combination of existing ideas (ICL and supervised learning) that addresses a specific limitation in current fine-tuning methods.

Problem Formulation: The authors identify a gap in the existing literature: the inconsistency between the behaviors of models during ICL and supervised fine-tuning. By framing the problem as an alignment issue, they provide a new perspective on how to improve instruction-following capabilities in large language models (LLMs).

Application to a New Domain: The paper demonstrates the effectiveness of IA2 in the context of instruction-following tasks, which is a critical domain for LLMs. This application is particularly relevant given the growing importance of LLMs in real-world scenarios where instruction following is essential.

**Weaknesses:**

- This aligns with the intermediate hidden states of knowledge distillation, logits alignment, and attention alignment, which seem to make little difference, limited only to different inputs?

- The experiments are conducted on a limited set of datasets and models. This may restrict the understanding of how IA2 performs across different model architectures and scales (<= 4B). For instruction-following tasks, larger and more challenging datasets such as SNI and P3 can be used.

- Compare IA2 with other alignment techniques such as prompt tuning, continuous prompt tuning, or other regularization methods like dropout or weight decay. This will help in understanding the unique benefits and trade-offs of IA2.

- The IA2 method involves additional computational overhead due to the alignment process. The paper does not discuss the computational efficiency or resource requirements of implementing IA2, which could be a significant limitation for practical applications.

- typos: line 257 `upto`

**Questions:**

see weakness

---

### Official Review · Reviewer_e95g · 2025-11-01

**Soundness:** 3
**Presentation:** 4
**Contribution:** 3
**Rating:** 6
**Confidence:** 4

**Summary:**

The paper proposed IA2 (ICL Activation Alignment), which employ a two stage training method to align the models activation (self-attention output) as when using ICL. The proposed method is based on the observation of the huge gap between models activation during SFT and ICL. The experiment results shows consistent improvement of the proposed IA2 over baseline (simple SFT) on a wide range of models and benchmarks.

**Strengths:**

1. The paper is well organized and easy to follow.
2. The observation of the gap of models activation between ICL and SFT is interesting, demonstrating different internal mechanisms of models when performing ICL and SFT.
3. The proposed IA2 method is simple and novel.

**Weaknesses:**

1. How the methods works on some of the SOTA open source model is unclear. While the authors show improvement on pre-trained models such as qwen3-4b-base, whether it can generalize to those post-training language models (eg. qwen3-4b) is unclear.
2. The authors tested the method on tasks with direct prediction without reasoning. While the performance on these tasks are good, recent LMs paradigm have been solving tasks such as gsm8k with reasoning process. It's unclear whether the IA2 method can help with this paradigm. One experiment I'll like to see is on gsm8k, compare sft and IA2 models performance when using reasoning during inference on qwen-4b-base model, to see if this method also help with models when performing reasoning.

**Questions:**

1. When IA2 aligns the activation (self-attention output) between ICL and SFT, I'm curious about the engineering choice. For example, it also makes sense to align attention weight instead of the attention output. Is there any specific reason that you choose to align the activations?
2. While IA2 shows performance improvement on some tasks that generally benefit from in-context examples, do you think it will work well on reasoning tasks that often does not benefit from in-context examples?

---

> ### Author Response · Authors · 2025-11-17
>
> **EDITED: Added TLDRs**
>
> Thank you for the clear and encouraging review. We are glad that you found the paper well-organized, the ICL–SFT activation gap interesting, and IA2 simple and novel. Regarding your doubts:
>
> > How the methods works on some of the SOTA open source model is unclear. …
>
> **TLDR: We have one experiment on instruct model (Llama-3.2-1B-Instruct). Our focus on base models (Line 123–125) was driven by **resource efficiency and controlled evaluation**.**
>
> Experimenting with post-trained models on _long/more difficult_ tasks requires a lot more resources and limits the extent of confounder testing (like rank/target modules/optimizer/reruns etc) we could do to claim statistically significant improvements. Using post-trained models also risks confounding results due to dataset contamination.
>
> However, IA2 is not limited to base models. In fact *our multi-token experiments already include an instruction tuned model* (Llama-3.2-1B-Instruct). We found no unusual trends with this model. In essence, IA2 distills the effect of a prompt into model weights. High activation similarity (Fig 2) shows that IA2 is quite effective in achieving this. The prompt could be ICL demos or a detailed instruction/other context.
>
> We agree that running IA2 on large post-trained models would be of interest; this is an exciting direction for future work that we will note in the updated draft.
>
>
> > One experiment I'll like to see is on gsm8k …
>
> **TLDR: Experiment already exists in multi-token segment. IA2 helps improve over SFT in this paradigm as well.**
>
> Our multi-token experiments already include exactly your requested setup, and we have results with GSM8K on the Qwen3-4B-base model. In line 243-245, we explain how the model performs reasoning before producing the answer (generates 200 tokens in total).
>
> In the multi-token case, we align activations over all output tokens (instead of only the last token in the single-token case). Our results (Table 2) show that IA2 does indeed help improve over SFT in this paradigm. If you have any followup questions, do let us know!
>
> We realize this may have been easy to miss in the draft, so we will explicitly highlight this in the revision.
>
> > … it also makes sense to align attention weight instead of the attention output …
>
> **TLDR: Base model weights are the same with or without ICL demos in the context.**
>
> We apologize if the following was not clear: in our target activation collection step for IA2, model weights remain *the same* (regardless of whether it sees [ICL demos + query] or only [query], base model weights do not change), so there is nothing to align in the weight space. It is because of attending different contexts, that the model produces different activations and outputs, which we can align.
>
> The objective of IA2 is to update model weights such that the activations produced by initial model weights on [ICL demos + query] are the same activations produced by the new model weights on just [query].
>
> > …do you think it will work well on reasoning tasks…
>
> **TLDR: We used chain-of-thought style prompting, which improves ICL performance for longer reasoning tasks. Our experiments include results on reasoning tasks. Our method improves over SFT in this setting as well.**
>
> For reasoning tasks we use chain-of-thought style in-context examples [1]. This helps to improve ICL performance on longer reasoning tasks (evident from the gap between w/o ICL and w/ ICL accuracy in Table 2). All our multi-token experiments (on GSM8K, HMathA, SciQ) are conducted under this setting. Moreover, we also show that our proposed method improves over SFT the same way it does in single-token prediction tasks (Table 2, 3). In our updated draft, we will make this detail clearer.
>
> NOTE: We will notify you once we have updated the paper draft with clarifications on reasoning tasks and chain-of-thought style ICL used for these tasks.
>
> ---
> References:
>
> [1] Chain-of-Thought Prompting Elicits Reasoning in Large Language Models, Jason Wei et al., 2022
>
> ---
>
> **We look forward to addressing any remaining questions during the discussion period and hope that our responses result in a higher evaluation score.**

---

> > ### Author Response · Authors · 2025-11-28
> >
> > Dear Reviewer,
> >
> > Thanks again for your valuable time reviewing our work.
> >
> > If our response has addressed your concern, we would appreciate an increased score. Otherwise, we are happy to engage with any remaining concerns.

---

### Official Review · Reviewer_uLVv · 2025-11-03

**Soundness:** 3
**Presentation:** 3
**Contribution:** 3
**Rating:** 6
**Confidence:** 3

**Summary:**

The paper proposes a two-stage fine-tuning method for low-resource settings to bridge the gap between task-specific fine-tuning and in-context learning: the method first performs a self-distillation step by matching the activation patterns in LLMs between the ICL prompt and the supervised fine-tuning (SFT) prompt, and then performs an SFT step upon the warm-started model. Experiments across multiple benchmarks show that the two-stage method outperforms SFT only and ICL only in low-data settings, with not only better accuracy, but also better calibration.

**Strengths:**

1. The method is well-motivated by the misalignment of the inner mechanism of transformers between ICL and SFT, and the proposed method is in principle well-designed to address such a misalignment by explicitly optimizing the exact objective.

2. The paper conducts solid experiments across multiple tasks, spanning both tasks with single-token outputs and multiple-token outputs, and demonstrates the empirical strength of the proposed method, not only in terms of accuracy, but also calibration.

3. The paper presents in-depth analyses in the discussion section and provides ablation results of different components of the proposed method.

**Weaknesses:**

1. Baselines from the context distillation literature are missing from the experiments. As the authors mentioned in the related work section, the proposed method is similar to context distillation methods in the literature, where models are trained to distill information in the context.  However, the proposed method is not compared against any appropriate context distillation methods, such as ICL distillation (Yukun Huang et al., 2022) and MetaICL (Sewon Min et al., 2022). The experimental results can be strengthened by including such baselines.

2. There is a potential unfair comparison in terms of data when comparing IA2 with ICL only and SFT only baselines. Given that IA2 only needs to run two passes of inference, one over the query-only data example and one over the query + demonstration data example, this gives IA2 effectively 2 times the data seen during training compared with ICL only and SFT only (ICL only sees the query+demonstration data example, and SFT only sees the query-only data example). Moreover, the paper focuses on the low-data regime, where such a misalignment of data sizes can lead to a large difference in performance. Thus, in my opinion, the proposed method needs to be compared with the baselines where the number of data examples is better aligned.

3. The proposed method does not perform consistently well on multi-token tasks, especially GSM8K and HMathA. Particularly in Table 2, SFT or IA2 is worse than the ICL-only baseline with 4 training examples. These results undermine the generalizability of the proposed method.

**Questions:**

See the weakness section.

**Details Of Ethics Concerns:**

1. Line 151: "100 test samples from many datasets": What exactly are the datasets used for evaluation here?

---

> ### Author Response · Authors · 2025-11-18
>
> Thank you for the constructive review, and for highlighting several strengths of our submission including the strong motivation grounded in the misalignment between ICL and SFT, the solid empirical evaluation across diverse tasks, and the depth of our analyses. We address your concerns below:
>
> > Baselines from the context distillation literature are missing …
>
> **TLDR: The setup of these baselines is incompatible with ours. While they train for better ICL during inference, we train to eliminate reliance on ICL.**
>
> We would like to point out that both Sewon Min et al, 2022, and Yukun Huang et al., 2022 _prime the base model to **better perform ICL** during inference_, i.e., they explicitly optimize the model to *rely on demonstrations*. In contrast, our objective is the opposite: *we eliminate the reliance on ICL at inference-time*! We do this by obtaining the qualities of ICL in SFT models (through IA2), thereby removing the need to use ICL during inference (and saving context space and compute). Because the inference-time assumptions differ, comparing IA2 to methods would not be apples-to-apples. We will make this distinction clearer in the revised draft.
>
> > IA2 only needs two passes of inference …
>
> **TLDR: All methods use the exact same data, and are allowed to see the data multiple times before training converges. Hence, the comparison is fair.**
>
> In our experiment setup:
> 1. *All* methods use the *exact same data* (Line 246-253), making the comparison information-theoretically fair.
> 2. *All* methods are allowed to train for up to 50 epochs, unless they converge earlier (decided by the dev set loss). So even SFT is allowed to use the data multiple times.
>
> Moreover, in context distillation literature, people typically use *teacher* models (often stronger models) to collect training signals. This requires a forward pass through the teacher. **Our method uses the same model for guidance**. The first pass is used to collect ICL’s training signal (activations/outputs). The second pass is where training happens, where the setup is identical to SFT. The richness of ICL and its utility in providing guidance for training is one of our main findings. We show that it is not “*how many* times you see the data” but “*how* you process the data” that matters.
>
> > The proposed method does not perform consistently well on multi-token tasks…
>
> **TLDR: ICL is more sample efficient for longer/more complex tasks. Key takeaway still holds: IA2->SFT consistently outperforms SFT only.**
>
> We appreciate that you noticed this subtle detail. Trained models are often unable to reach ICL performance for low shot counts in multi-token settings. In the table below, we show the evolution of model performance with increasing shot count [acc-mean (acc-std)]. We also show the gap in performance of IA2->SFT wrt to ICL and SFT in the last two columns [difference in means].
>
> | Dataset     | ICL         | SFT         | IA2         | IA2 -> SFT  | Gap wrt ICL | Gap wrt SFT |
> |-------------|-------------|-------------|-------------|-------------|-------------|-------------|
> | GSM8K, N=2  | 81.2 (00.4) | 65.2 (01.0) | 70.7 (01.8) | 74.6 (02.4) | -06.6       | +09.4       |
> | GSM8K, N=4  | 76.4 (01.1) | 70.9 (02.8) | 77.4 (02.8) | 73.6 (03.7) | -02.8       | +02.7       |
> | GSM8K, N=8  | 77.2 (00.9) | 73.5 (01.6) | 78.8 (02.5) | 76.0 (02.0) | -01.2       | +02.5       |
> | HMathA, N=2 | 58.6 (02.0) | 47.8 (04.0) | 33.9 (06.4) | 48.7 (07.0) | -09.9       | +00.9       |
> | HMathA, N=4 | 60.4 (02.1) | 50.4 (01.8) | 47.8 (06.0) | 55.3 (02.1) | -05.1       | +04.9       |
> | HMathA, N=8 | 62.1 (01.0) | 52.0 (00.8) | 57.3 (03.7) | 59.4 (00.8) | -02.7       | +07.4       |
>
> Key takeaways from this table:
>
> **(1)**: ICL shows more sample efficiency than all other training methods.
>
> **(2)**: IA2->SFT quickly closes the gap in performance wrt ICL with increasing shot count.
>
> **(3)**: IA2->SFT maintains a lead over SFT consistently (which is our proposed enhancement).
>
> Additionally, the Qwen model went through [mid-training](https://huggingface.co/Qwen/Qwen3-4B-Base) on STEM data. This can cause some weird behavior (like ICL performance at N=2 is higher than at N=4,8 for GSM8K) and possibly confound some results. We will update the paper with a discussion of this detail.
>
> > Line 151: "100 test samples from many datasets": What exactly are the datasets used for evaluation here?
>
> This detail is already provided in Appendix B (Line 887-888, linked at line 157). To reiterate, we used all datasets in our paper (single/multi-token) to prepare Figure 2. We agree that saying “many datasets” makes this unclear, and will fix it in the updated version of the paper.
>
> NOTE: We will notify once the draft is updated with the following changes: discussion of ICL performance > trained models, and clarity around Figure 2.
>
> ---
>
> **We look forward to addressing any remaining questions during the discussion period and hope that our responses result in a higher evaluation score.**

---

> > ### Author Response · Authors · 2025-11-28
> >
> > Dear Reviewer,
> >
> > Thanks again for your valuable time reviewing our work.
> >
> > If our response has addressed your concern, we would appreciate an increased score. Otherwise, we are happy to engage with any remaining concerns.

---

### Official Review · Reviewer_1ePT · 2025-11-04

**Soundness:** 3
**Presentation:** 3
**Contribution:** 3
**Rating:** 6
**Confidence:** 4

**Summary:**

The paper IA2 first aligns a model’s internal activations to those it produces during in-context learning, then runs standard supervised fine-tuning—yielding SFT models that better mimic ICL and show improved accuracy and calibration across tasks in low-data settings.

**Strengths:**

- IA2 offers a simple, data-efficient, label-free priming signal (activation matching) that can precede ordinary SFT.

- This paper has a broad empirical sweep with many tasks (SST-2/FinPhrase/PoemSent, AGNews/BBCN, StrategyQA, SciQ, GSM8K, MATH-Alg), two model families, varied #examples; separate 500-example eval sets held out from all adaptation methods.

**Weaknesses:**

1. LoRA edits only WQ, WK, WO (not WV), rank fixed at 8; authors note selections “subject to improvement,” but this restriction could cap IA2 capacity and bias results vs SFT-only. Ablations on rank/targets would clarify.

2. Did the authors consider direct comparisons to context distillation or representation-distillation baselines that imitate ICL outputs (e.g., teacher-student on logits/hidden states) under the same setup? I didn't see any results reported in the main results; that would better isolate IA2’s contribution beyond output-level distillation. (Related work, e.g., [1]-[4] is cited but not reproduced head-to-head.)

---

[1] Gustavo Aguilar et al., Knowledge distillation from internal representations, 2020.

[2] Tong Chen et al., Demonstration distillation for efficient in-context learning, 2024.

[3] Lucas Caccia et al., Training plug-n-play knowledge modules with deep context distillation, 2025.

[4] Brian K. Chen et al., Exact conversion of in-context learning to model weights in linearized-attention transformers, 2024.

**Questions:**

See weaknesses

---

> ### Author Response · Authors · 2025-11-17
>
> Thank you for your thoughtful and constructive review, and for highlighting both the simplicity of IA2, and the empirical breadth of our experiments! We address your concerns below:
>
> > LoRA edits only WQ, WK, WO (not WV), rank fixed at 8; …
>
> **TLDR: We present ablations and find that these variables do not impact our claim—IA2 -> SFT consistently outperforms SFT only.**
>
> Thanks for bringing this up. Our note (“subject to improvement”) was based on preliminary ablations on rank/targets (rank [8,16]; and [QKV, QKO, QK] on some datasets) to make sure our results were not biased against SFT only. **As our choices were consistent, no training method has an unfair advantage.** We are sharing a slice of these ablations below for Llama 3.2-1B models trained on the SST2 dataset (evaluated on 3 datasets) and Qwen3-4B-base models on the GSM8K dataset (evaluated on itself). Each numerical column contains [acc-mean (acc-std), ece-mean (ece-std)] across 5 re-runs for the best performing hyperparameters (learning rate, beta).
>
> | Dataset (lora params) | ICL                      | SFT                      | IA2                      | IA2->SFT                 |
> |-----------------------|--------------------------|--------------------------|--------------------------|--------------------------|
> | FinS (r8, qko)        | 60.2 (03.0), 0.10 (0.02) | 60.0 (13.6), 0.33 (0.11) | 53.2 (20.4), 0.33 (0.04) | 89.8 (09.0), 0.09 (0.08) |
> | PoemS (r8, qko)       | 57.5 (01.8), 0.11 (0.02) | 50.7 (03.7), 0.41 (0.07) | 61.2 (12.5), 0.22 (0.06) | 84.8 (08.1), 0.14 (0.08) |
> | SST2 (r8, qko)        | 77.0 (02.2), 0.10 (0.02) | 53.5 (05.0), 0.34 (0.11) | 64.9 (15.3), 0.30 (0.05) | 90.4 (01.3), 0.09 (0.02) |
> | FinS (r8, qk)         | 60.2 (03.0), 0.10 (0.02) | 48.8 (16.1), 0.21 (0.12) | 51.9 (12.6), 0.14 (0.02) | 50.4 (21.6), 0.27 (0.07) |
> | PoemS (r8, qk)        | 57.5 (01.8), 0.11 (0.02) | 48.9 (03.1), 0.24 (0.14) | 42.6 (06.2), 0.13 (0.04) | 56.7 (12.7), 0.29 (0.13) |
> | SST2 (r8, qk)         | 77.0 (02.2), 0.10 (0.02) | 50.0 (06.4), 0.33 (0.13) | 61.6 (08.3), 0.17 (0.09) | 65.7 (15.0), 0.27 (0.14) |
> | FinS (r8, qkv)        | 60.2 (03.0), 0.10 (0.02) | 54.2 (15.8), 0.34 (0.16) | 66.2 (01.9), 0.16 (0.06) | 87.3 (05.4), 0.06 (0.02) |
> | PoemS (r8, qkv)       | 57.5 (01.8), 0.11 (0.02) | 53.4 (05.5), 0.36 (0.11) | 57.4 (08.0), 0.20 (0.08) | 80.0 (07.2), 0.19 (0.07) |
> | SST2 (r8, qkv)        | 77.0 (02.2), 0.10 (0.02) | 53.5 (03.4), 0.34 (0.08) | 65.3 (18.6), 0.34 (0.03) | 88.7 (02.8), 0.10 (0.03) |
> | FinS (r16, qko)       | 60.2 (03.0), 0.10 (0.02) | 55.4 (13.0), 0.25 (0.12) | 51.7 (19.4), 0.33 (0.04) | 86.8 (09.4), 0.13 (0.09) |
> | PoemS (r16, qko)      | 57.5 (01.8), 0.11 (0.02) | 54.5 (10.9), 0.26 (0.16) | 57.2 (09.1), 0.27 (0.06) | 79.5 (10.0), 0.13 (0.05) |
> | SST2 (r16, qko)       | 77.0 (02.2), 0.10 (0.02) | 55.8 (13.9), 0.29 (0.14) | 65.5 (14.9), 0.30 (0.05) | 87.7 (05.8), 0.12 (0.06) |
> | GSM8K (r8, qko)       | 76.4 (01.1)              | 70.9 (02.8)              | 77.4 (02.8)              | 73.6 (03.7)              |
> | GSM8K (r16, qko)       | 76.4 (01.1)              | 70.3 (05.5)              | 77.8 (06.4)              | 74.1 (03.3)              |
>
>
>
> **Takeaways:**
>
> **(1)** The choice of QKO [vs QKV] does not impact our main finding: IA2 -> SFT is much more performant than SFT only.
>
> **(2)** Similarly, lora rank 8 -> 16 does not impact the results much.
>
> **(3)** Lastly, targeting only QK matrices (as these are used for cross-token reference), does reduce all performance numbers drastically, but still maintains a healthy gap between SFT only and IA2 -> SFT.

---

> > ### Author Response · Authors · 2025-11-17
> >
> > > … direct comparisons to context distillation or representation-distillation baselines …
> >
> > **TLDR: Existing approaches differ in setting and assumptions. We present new experimental results to show that our work does not compete with rather complement prior work.**
> >
> > We want to highlight some key differences between our method and these baselines.
> > 1. [1, 3] use a stronger teacher model for training signals through output/representation distillation. While useful, it is not always possible as more frontier models begin restricting access [[news](https://x.com/maksym_andr/status/1989770016607088742?s=20)]. In contrast, **our method uses the same model for guidance, through its own ICL**. The richness of ICL and its utility in providing guidance for training is one of our main findings. Our experimental results in Table 3, 6 use the **same**-model-with-ICL as the teacher model, aligning with these baselines. Hence, **our work is a complement and not in competition with existing context distillation methods**. To further support this claim, we implement a KD loss baseline [1], with soft labels collected under ICL from the same model. The results are discussed after this list.
> > 2. [3] distills relevant documents and other context (in a continued pre-training fashion) to better answer queries related to it. In our work, we only use paired training-samples (given X, output Y) to understand a task, making our setups incompatible.
> > 3. [2] uses LLM agents to compress context length *but still rely on context*  during inference. In contrast, IA2 removes the context (demonstrations) completely at inference time to save context space and compute. Again, this makes our setups incompatible.
> > 4. We actually take inspiration from [4] (line 187-188) in trying to exactly distill the effect of a prompt into model weights. But we **extend it** to a more practical version instead of using approximate softmax kernels, overcoming scalability issues. As the authors themselves note: “Regarding the approximate softmax conversion method, scalability is an issue.”
> >
> >
> > **KD-loss baseline experiment**: To directly address your question, we trained Llama-3.2-1B-Instruct for GSM8K and Llama-3.2-1B for SST2, with soft labels collected under ICL from the *same* model, and evaluated the same dataset (i.e., ID evaluation). Other experiment conditions are kept identical to our setup in the paper. In the table below, each numerical column contains [acc-mean (acc-std), ece-mean (ece-std) {only for SST2}] like the previous table.
> >
> > | Dataset     | SFT (on ICL outputs)   | KD (wrt ICL)             | IA2                      | IA2 -> SFT               | IA2 + SFT                |
> > |-------------|------------------------|--------------------------|--------------------------|--------------------------|--------------------------|
> > | GSM8K (N=4) | 28.2 (04.5)            | 36.1 (03.5)              | 29.3 (05.0)              | 30.3 (03.2)              | **37.0** (01.9)              |
> > | SST2 (N=8)  | 52.5 (04.9), 0.31 (00.1) | 55.9 (08.8), 0.16 (0.06) | 64.9 (15.3), 0.30 (0.05) | 64.7 (19.3), 0.33 (0.19) | **66.9** (18.4), **0.28** (0.21) |
> >
> > **Observations**: Although KD loss increases the performance in the multi-token case (possibly due to denser signal from multiple tokens) almost equalling IA2 + SFT, it does not perform well on the single token task, where IA2 based training is better. Overall, these new results further strengthen our proposition of self-guidance through ICL.
> >
> >
> > NOTE: In the coming days, we will update the draft (and notify you) with the following additions:
> > 1. Ablation study results and discuss why our experiment choices are justified.
> > 2. The KD-loss baseline experiment.
> >
> > ---
> >
> > References:
> >
> > [1] Gustavo Aguilar et al., Knowledge distillation from internal representations, 2020.
> >
> > [2] Tong Chen et al., Demonstration distillation for efficient in-context learning, 2024.
> >
> > [3] Lucas Caccia et al., Training plug-n-play knowledge modules with deep context distillation, 2025.
> >
> > [4] Brian K. Chen et al., Exact conversion of in-context learning to model weights in linearized-attention transformers, 2024.
> >
> > ---
> >
> > **We look forward to addressing any remaining questions during the discussion period and hope that our responses result in a higher evaluation score.**

---

> > > ### Author Response · Authors · 2025-11-28
> > >
> > > Dear Reviewer,
> > >
> > > Thanks again for your valuable time reviewing our work.
> > >
> > > If our response has addressed your concern, we would appreciate an increased score. Otherwise, we are happy to engage with any remaining concerns.

---

### Meta-Review · Area_Chair_J71p · 2026-01-07

**Summary:**

The paper shows that In-Context Learning (ICL) and Supervised Fine-Tuning (SFT) rely on different internal activation patterns, which explains why ICL often generalizes and calibrates better in low-data settings.

**Reviewer Concerns:**

- head-to-head comparisons against context/distillation variants

- restricting LoRA to certain attention matrices (and fixing rank) could bias outcomes

- Fairness of data/compute comparison

- inconsistent gains on multi-token/reasoning tasks

- generalization to post-trained/instruct models unclear

**Reviewer Scores:**

The reviewers were unanimously positive about the paper, and their scores might raise further post discussion.

---

### Decision · Program_Chairs · 2026-01-26

Accept (Poster)